# Three stepwise pH progressions in stratum corneum for homeostatic maintenance of the skin

Keitaro Fukuda[1,2,8], Yoshihiro Ito [2,8], Yuki Furuichi[1,2,8], Takeshi Matsui [1,2,3], Hiroto Horikawa[1,2], Takuya Miyano [4], Takaharu Okada [5,6], Mark van Logtestijn[4], Reiko J. Tanaka [4], Atsushi Miyawaki [7] & Masayuki Amagai [1,2] ✉

The stratum corneum is the outermost skin layer with a vital role in skin barrier function. It is comprised of dead keratinocytes (corneocytes) and is known to maintain its thickness by shedding cells, although, the precise mechanisms that safeguard stratum corneum maturation and homeostasis remain unclear. Previous ex vivo studies have suggested a neutral-to-acidic pH gradient in the stratum corneum. Here, we use intravital pH imaging at single-corneocyte resolution to demonstrate that corneocytes actually undergo differentiation to develop three distinct zones in the stratum corneum, each with a distinct pH value. We identified a moderately acidic lower, an acidic middle, and a pH-neutral upper layer in the stratum corneum, with tight junctions playing a key role in their development. The upper pH neutral zone can adjust its pH according to the external environment and has a neutral pH under steady-state conditions owing to the influence of skin microbiota. The middle acidic pH zone provides a defensive barrier against pathogens. With mathematical modeling, we demonstrate the controlled protease activation of kallikrein-related peptidases on the stratum corneum surface that results in proper corneocyte shedding in desquamation. This work adds crucial information to our understanding of how stratum corneum homeostasis is maintained.

Multicellular organisms depend on surface barriers to compartmentalize themselves from harsh extreme environments and maintain homeostasis[1]. In terrestrial vertebrates, the body surface is sealed by the stratified squamous epithelium of skin called epidermis[2]. The transition from aquatic to terrestrial life in vertebrates caused selective pressures affecting epidermal anatomy, and terrestrial vertebrates developed an air-liquid interface barrier, known as the stratum corneum (SC), as the outermost layer of their epidermis (Fig. 1a)[3,4]. The SC

possesses unique characteristics, as it prevents the entry of pathogens and allergens (outside-in barrier), as well as the leakage of water (inside-out barrier), although it consists of layers of nonviable enucleated keratinocytes (corneocytes). It can also maintain site-specific thickness by shedding scales (desquamation) every day. Furthermore, the SC hosts the skin microbiota that plays essential roles in priming the innate and adaptive skin immune systems[5]. These characteristics are a result of a mechanism that maintains homeostasis in the SC.

[1]Laboratory for Skin Homeostasis, RIKEN Center for Integrative Medical Sciences, Kanagawa, Japan. [2]Department of Dermatology, Keio University School of Medicine, Tokyo, Japan. [3]Laboratory for Evolutionary Cell Biology of the Skin, School of Bioscience and Biotechnology, Tokyo University of Technology, Tokyo, Japan. [4]Department of Bioengineering, Imperial College London, London, UK. [5]Laboratory for Tissue Dynamics, RIKEN Center for Integrative Medical Sciences, Kanagawa, Japan. [6]Graduate School of Medical Life Science, Yokohama City University, Kanagawa, Japan. [7]Laboratory for Cell Function Dynamics, RIKEN Center for Brain Science, Saitama, Japan. [8]These authors contributed equally: Keitaro Fukuda, Yoshihiro Ito, Yuki Furuichi. ✉e-mail: amagai@keio.jp

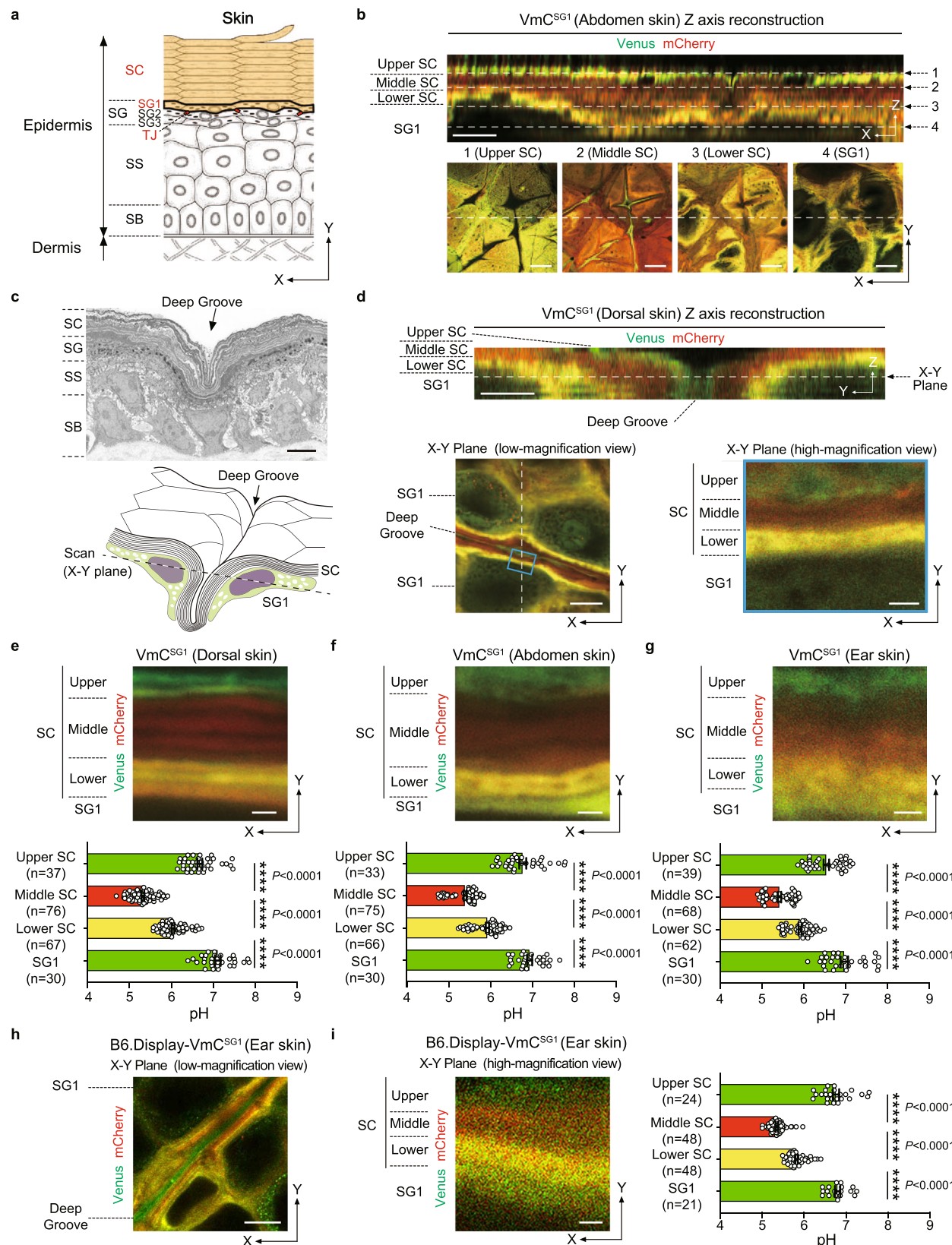

However, the underlying mechanisms safeguarding SC homeostasis remain elusive.

The corneocytes are linked together by corneodesmosomes, and their intercellular spaces are sealed with lipids. The intracellular spaces of corneocytes are filled with proteins such as keratin filaments and filaggrin, along with their degradation products[6,7]. The SC contains enzymes responsible for lipid synthesis and protein degradation, and the catalytic activities of these enzymes are regulated by pH[8,9]. In addition, corneocytes are generated through a unique mode of cell death of the uppermost stratum granulosum keratinocytes (SG1 cells, Fig. 1a) called corneoptosis[10]. It requires prolonged intracellular $Ca^{2+}$ elevation (phase I) followed by controlled intercellular acidification

**Fig. 1 | Three stepwise SC-pH zones preserved across various body parts.**
**a** Structure of the epidermis. The epidermis is composed of the stratum basale (SB), stratum spinosum (SS), stratum granulosum (SG), and stratum corneum (SC). The SG is composed of SG1, SG2, and SG3. Tight junctions (TJs, red) are formed between adjacent SG2 cells. The present study examined the pH distribution and pH values within the SC and SG1 (orange). **b** Representative confocal images of the upper epidermis of abdomen skin of VmC$^{SG1}$ mice, as observed under Z-sliced reconstituted (top) and high-magnification X-Y plane images (bottom). Scale bars, 5 (top), 10 (bottom) μm. **c** Representative electron microscopy image (top) and scheme of X-Y plane confocal image (bottom) of the SC groove. Scale bar, 5 μm. **d** Representative confocal images of the upper epidermis of dorsal skin of VmC$^{SG1}$ mice, as observed under Z-sliced reconstituted images (top) and low (bottom-left) and high (bottom-right, region marked by blue lines in left) magnification X-Y plane. Scale bars, 5 (top), 10 (bottom-left), and 1 (bottom-right) μm. High-

magnification X-Y plane confocal images of SC-pH zones (top) of the **e** dorsal, **f** abdominal, and **g** ear skin of VmC$^{SG1}$ mice. pH values in each SC-pH zone and SG1 (bottom) of the **e** dorsal ($n$ = 37, 76, 67, and 30 spots from four biologically independent animals, respectively), **f** abdominal ($n$ = 33, 75, 66, and 30 spots from four biologically independent animals, respectively), and **g** ear ($n$ = 39, 68, 62, and 30 spots from four biologically independent animals, respectively) skin. Scale bar, 1 μm. Representative **h** low and **i** high-magnification X-Y plane confocal images of the ear skin in B6.Display-VmC$^{SG1}$ mice (left). pH values in each SC-pH zone and SG1 (right, $n$ = 24, 48, 48, and 21 spots from three biologically independent animals, respectively). Scale bar, 10 μm (**h**) and 1 μm (**i**). Data are shown as mean ± SEM and are pooled from four (**e**–**g**) or three experiments (**i**) or are representative of at least three independent experiments (**b**–**d** and **h**). ****$p$ < 0.0001; Kruskal-Wallis one-way ANOVA followed by Dunn's post hoc test (**e**–**h**). Source data are provided as a Source Data file.

(phase II), thereby activating DNases and proteases to eliminate organelles such as mitochondria without causing inflammation[11]. Based on these results, we hypothesized that SC-pH affects SC homeostasis.

Previous studies have employed two main methods for measuring the pH distribution of human SC: applying a pH electrode to the skin surface before and after consecutive tape stripping or performing fluorescence lifetime imaging of three-dimensional human skin constructs stained with a pH indicator dissolved in organic solvents using two-photon scanning microscopy[12–14]. The SC-pH profile has been reported to exhibit a monotonous gradient, ranging from pH 6.8–7.2 at the stratum granulosum (SG; the uppermost viable keratinocyte cell layer that is located beneath the SC) to pH 4.5–5.5 at the SC surface[12–14]. However, tape stripping is known to induce skin inflammation[15,16], and the organic solvents may dissolve lipids surrounding corneocytes, which both can affect SC-pH. Thus, previously reported SC-pH measurements may not reflect steady-state in vivo SC-pH values and their distribution. Indeed, kallikrein-related peptidase (KLKs) that initiate desquamation remain inactive at acidic pH and this could not clearly explain why KLKs execute desquamation in the upper SC[17,18]. Furthermore, owing to the presence of skin microbiota on the SC surface[19,20], it is unlikely that the SC is acidic in vivo, as this pH range is less favorable for commensal bacterial colonization than neutral pH values[21].

Here, we performed quantitative intravital pH imaging in corneocytes using transgenic mice expressing a ratiometric pH indicator, considering the nuanced SC-pH distribution. We aimed to elucidate the in vivo SC-pH distribution and biological significance of pH in SC homeostasis. Studies that used earlier SC-pH measurements have suggested that the SC possesses a monotonous neutral-to-acidic gradient profile and that acidic microenvironment promotes skin antimicrobial defense[12–14,17,18]. However, in this study, we found that corneocytes undergo differentiation, thereby inducing three-tiered pH zonation; namely, this process generates the lower-moderately acidic (pH 6.0), middle-acidic (pH 5.4), and upper-nearly neutral (pH 6.7) zones. The lower SC-pH zone originates from the acidification during corneoptosis phase II. The middle SC-pH zone is the acid mantle layer that serves as a protective barrier against pathogens. The upper SC-pH zone is surprisingly adaptive; it can adjust its pH based on external conditions, exhibiting a near-neutral pH under steady-state owing to skin microbiota. The three-tiered pH zones are suitable for regulating protease catalytic activity to induce desquamation on the SC surface. Our results establish that SC is not merely a layer of dead keratinocytes. Instead, the SC comprises three distinct pH zones contributing to SC differentiation, thereby maintaining SC homeostasis.

## Results
### Generation of genetically engineered mice for the measurement and visualization of corneocyte pH
To visualize the in vivo SC-pH distribution and values, we generated fusion proteins using different pH-sensitivity probes. We used Venus

or VenusH148G as the high pH-sensitive probes and mCherry as the low pH-sensitive probe[10,22,23]. We generated hairless (HR) mice that specifically expressed these fusion proteins under the skin aspartic protease (SASP) locus, thereby expressing these pH probes in the SG1 cell layer (HR.*SASP*$^{Venus-mCherry/+}$ [VmC$^{SG1}$] and HR.*SASP*$^{VenusH148G-mCherry/+}$ [VH148GmC$^{SG1}$] mice, respectively [Supplementary Fig. 1a]). Therefore, these mice are identical to the previously described SASP heterozygous knockout mouse[24], which does not have a phenotype, except that the integration of the Veuns-mcherry (VmC) or Venus$^{H148G}$-mCherry (VH148GmC) cassette into the SASP locus. To assess whether the fluorescence of Venus-mCherry (VmC) and Venus$^{H148G}$–mCherry (VH148GmC) proteins, expressed in the murine SC, reflected the pH changes, we isolated SG1 cells from the dorsal skin of these mice and exposed them into various pH buffers (pH 4.1–8.0) with protonophores (nigericin and valinomycin) that equilibrated the intracellular pH to the buffer pH. The fluorescence intensities of Venus, VenusH148G, and mCherry at each pH value were analyzed using confocal microscopy (Supplementary Fig. 1b–e). The differences in the Venus, VenusH148G, and mCherry fluorescence profiles in response to pH variations produced specific ratiometric curves for Venus/mCherry and VenusH148G/mCherry fluorescence (Supplementary Fig. 1f, g). When the Venus/mCherry fluorescence ratio was normalized to the corresponding value at pH 8.0 (relative Venus/mCherry), the ratiometric curves of the relative Venus/mCherry fluorescence measured under two laser powers overlapped (Supplementary Fig. 1f). After normalization, the ratiometric curves of VenusH148G/mCherry measured under two laser powers also overlapped (Supplementary Fig. 1g). These results suggest that the pH of corneocytes and SG1 cells in both VmC$^{SG1}$ and VH148GmC$^{SG1}$ mice could be calculated by substituting the relative Venus/mCherry and Venus$^{H148G}$/mCherry values, respectively, into the ratiometric curves established using isolated SG1 cells. Furthermore, the pKa values of VmC and VH148GmC were approximately 5.8 and 7.0, respectively (Supplementary Fig. 1f, g), suggesting that VmC is more suitable for visualizing pH changes in acidic conditions than VH148GmC.

### SC comprises three stepwise pH zones across various body parts
To observe the SC-pH distribution in mice in vivo, we performed intravital imaging of VmC$^{SG1}$ and VH148GmC$^{SG1}$ mice via confocal microscopy (Fig. 1a). The reconstituted Z-sliced section revealed that the SC of dorsal skin comprised three-tiered zonation: upper-nearly neutral, middle-acidic, and lower-moderately acidic. However, resolving the pH in single corneocytes was difficult (Fig. 1b and Supplementary Movie 1). As the skin exhibited grooved wrinkles (Fig. 1c) and the confocal microscopic image in the X-Y plane offered higher resolution than the reconstituted Z-sliced section (Fig. 1b), we focused on the grooved wrinkles in the X-Y plane image (Fig. 1c). This method enabled us to visualize the SC-pH at the single-corneocyte resolution (Fig. 1d, e). In addition to the dorsal skin, the abdomen, ear, tail, and palm skin of mice exhibited a three-tiered zonation (Fig. 1f, g, and Supplementary Fig. 2a–e).

Subsequently, we calculated the pH value of each zone in the SC of VmC$^{SG1}$ mice. The upper, middle, and lower SC-pH zones and the SG1 cell layer had significantly different pH values, which ranged between 6.58–6.77, 5.32–5.42, 5.92–6.04, and 6.93–7.11, respectively (Fig. 1e–g). Collectively, the SC had three stepwise SC-pH zones, which were maintained across various body sites.

Given that VmC$^{SG1}$ and VH148GmC$^{SG1}$ mice enabled intravital imaging of the intercellular pH of corneocytes and exhibited three-tiered zonation, we speculated whether the extracellular pH of corneocytes shows similar pH changes. To investigate this, we generated B6 mice that specifically expressed Venus-mCherry fusion protein on the extracellular side of the plasma membrane of SG1 cells and corneocytes (B6.*SASP*$^{Display-Venus-mCherry/+}$ mice [B6. Display-VmC$^{SG1}$ mice]) and performed intravital SC-pH imaging. Similar to VmC$^{SG1}$ and VH148GmC$^{SG1}$ mice, B6. Display-VmC$^{SG1}$ mice exhibited three-tiered zonation (Fig. 1h and Supplementary Movie 2). The upper, middle, and lower SC-pH zones and the SG1 cell layer had significantly different pH values [pH 6.76 (95% CI 6.61–6.91), 5.36 (95% CI 5.30–5.41), 5.82 (95% CI 5.76–5.88), and 6.80 (95% CI 6.67–6.93), respectively] (Supplementary Fig. 2h). These results suggest that the extracellular pH of corneocytes matches with the intracellular pH of corneocytes.

As the fluorescence of Venus can be quenched by pH as well as by the chloride ion concentration ([Cl$^-$]) while the fluorescence of EGFP cannot be quenched by [Cl$^-$][25], we performed intravital imaging of HR mice expressing EGFP in SG1 cells (EGFP$^{SG1}$ mice). Similar to VmC$^{SG1}$ mice, EGFP$^{SG1}$ mice possessed three distinct zones of EGFP intensity in the SC−upper-high EGFP, middle-low EGFP, and lower-high EGFP (Supplementary Fig. 3a)−indicating that the quenched Venus fluorescence was not due to [Cl$^-$] changes. Furthermore, we detected EGFP protein expression in EGFP$^{SG1}$ mice and Venus and mCherry expression in VmC$^{SG1}$ mice via western blot analyses of SC samples obtained from tape stripping (TS). The amounts of EGFP, Venus, and mCherry were similar from the 1$^{st}$ (upper SC) to the 9$^{th}$ (lower SC) TS samples; however, the level of filaggrin decreased toward the SC surface, consistent with earlier observations (Supplementary Fig. 3b)[26]. These results suggest that the changes in Venus-mCherry fluorescence within the SC were not due to proteasomal degradation but because of pH-induced quenching in the SC.

### The lower SC-pH zone originates from acidification during corneoptosis, and the acidification of the three stepwise SC-pH zones depends on tight junctions

Filaggrin is known to exist within the lower SC region of corneocytes and subsequently degrade into natural moisturizing factors, thereby playing a vital role in skin hydration[26–28]. To examine the distribution of filaggrin within the SC-pH zones, we performed immunofluorescence staining of filaggrin in VmC$^{SG1}$ mouse skin. Filaggrin was observed in the cell membrane of corneocytes located in the lower SC-pH zone and disappeared above the lower SC-pH zone (Fig. 2a). This result suggests that the lower SC-pH zone corresponds to the previously reported lower SC region, where filaggrin exists[26], and that filaggrin degrade into natural moisturizing factors at the middle SC-pH zone.

As the corneocytes are formed through SG1 cell death (corneoptosis) and SG1 cells become acidic during corneoptosis phase II[10], we next aimed to determine the location and pH value of SG1 cells in corneoptosis phase II. To do this, we injected Hoechst 33242 into the skin of VmC$^{SG1}$ and VH148GmC$^{SG1}$ mice and performed intravital SC-pH imaging. The SG1 cells with yellow cytoplasm and Hoechst 33242 nuclear staining (which corresponds to corneoptosis phase II) were localized to the lowermost layer of the lower SC-pH zone (Fig. 2b, c, and Supplementary Fig. 3c, and Supplementary Movie 3). The pH of SG1 cells in corneoptosis phase II was similar to that of corneocytes in the lower SC-pH zone (Fig. 2d), suggesting that the lower SC-pH zone originates from the acidification that occurs during corneoptosis phase II.

In mammalian skin, the epidermis has two types of physical barriers: tight junctions (TJs) and the SC. TJs are formed at the apical side of keratinocytes at the second layer of the SG (Fig. 1a) and are in contact with SG1 cells to function as liquid−liquid interface barriers[29,30]. Mice lacking claudin-1 (*Cldn1*$^{-/-}$), a four-transmembrane protein that forms TJ proteins, are known to possess a more tightly compacted and thicker SC, and experience higher water loss through the epidermis than wild-type mice[31]. Claudin-1 deficient patients are known to develop ichthyosis[32], an inherited keratotic disorder characterized by impaired SC differentiation and thickening owning to impaired desquamation. Furthermore, patients with atopic dermatitis (AD), a chronic inflammatory skin disease leading to high surface SC-pH values[16], have significantly low claudin-1 protein expression in their epidermis[33]. Based on these reports, we hypothesized that the distributions and values of SC-pH may be abnormal in *Cldn1*$^{-/-}$ mice. As VH148GmC$^{SG1}$ mice have a higher sensitivity to neutral pH values than VmC$^{SG1}$ mice, and *Cldn1*$^{-/-}$ mice die within one day of birth (owing to excessive epidermal water loss)[31], we examined the SC-pH of neonatal *Cldn1*$^{-/-}$ VH148GmC$^{SG1}$ mice. We observed the absence of the three stepwise SC-pH zones, and the entire SC exhibited a neutral pH (Fig. 2e, f), indicating that the acidification of the lower and middle SC-pH zones requires intact TJs. Collectively these results suggest that the lower SC-pH zone is a foundation layer of three stepwise SC-pH zones and is hampered by TJ disruption.

Furthermore, we examined the SC-pH of epidermis-specific *Cldn1*$^{-/-}$ VH148GmC$^{SG1}$ mice (B6.*Cldn1*$^{fl/fl}$ *K14*-CreERT VH148GmC$^{SG1}$ mice). Similar to neonatal *Cldn1*$^{-/-}$ VH148GmC$^{SG1}$ mice, the entire SC of adult B6.*Cldn1*$^{fl/fl}$ *K14*-CreERT VH148GmC$^{SG1}$ mice exhibited a neutral pH and were thicker compared with that of VmC$^{SG1}$ mice (Supplementary Fig. 3d, e). Surprisingly, immunofluorescence staining of filaggrin in B6.*Cldn1*$^{fl/fl}$ *K14*-CreERT VH148GmC$^{SG1}$ mice skin revealed that filaggrin distributed with accumulation along with the cell membrane of corneocytes was located in the entire SC (Fig. 2g). However, filaggrin monomer was undetectable in B6.*Cldn1*$^{fl/fl}$ *K14*-CreERT VH148GmC$^{SG1}$ mice, as confirmed by western blot analyses of SC samples obtained from TS (Supplementary Fig. 3f). Instead, bands approximately the size of dimetric, trimetric, and tetrameric filaggrin were detected. Collectively, these results suggest that *Cldn1* deficiency resulted in the loss of three stepwise SC-pH zones and the accumulation of prematurely processed filaggrin. These TJ-impaired conditions are considered to result in decreased filaggrin monomer and suppression of subsequent natural moisturizing factor production, which would lead to impaired desquamation.

### The upper SC-pH zone has an acclimation-like property and is susceptible to external pH changes

Given that the middle-acidic SC-pH zone further differentiates into the upper-nearly neutral SC-pH zone, we examined the impact of external stimuli on pH zone differentiation. We applied STELLA Fluor 650 (a fluorescent dye whose fluorescence intensity is stable at pH 3.0–10.0; dissolved in phosphate-buffered saline) containing Milli-Q water to the dorsal skin surface of VmC$^{SG1}$ mice. STELLA Fluor 650 signals were only observed in the upper SC layer, which corresponded to the upper SC-pH zone (Fig. 3a). This result was consistent with previous observations based on time-of-flight secondary ion mass spectrometry (TOF-SIMS), which suggested the presence of a surface zone that allowed passive ion (i.e., potassium [K$^+$]) influx and efflux[28]. Furthermore, we applied buffers with various pH values (pH 4.1–7.4) to the dorsal and abdominal skin of VmC$^{SG1}$ mice and measured the pH of each SC-pH zone and SG1. In both the dorsal and abdominal skin, the pH of the upper SC-pH zone significantly changed after the buffer treatment. In contrast, the middle and lower SC-pH zones presented pH values similar to those in untreated mice (Fig. 3b, c). Therefore, the upper SC-pH zone exhibits acclimation-like properties and can adjust its pH according to external pH changes, unlike the lower and middle SC-pH zones.

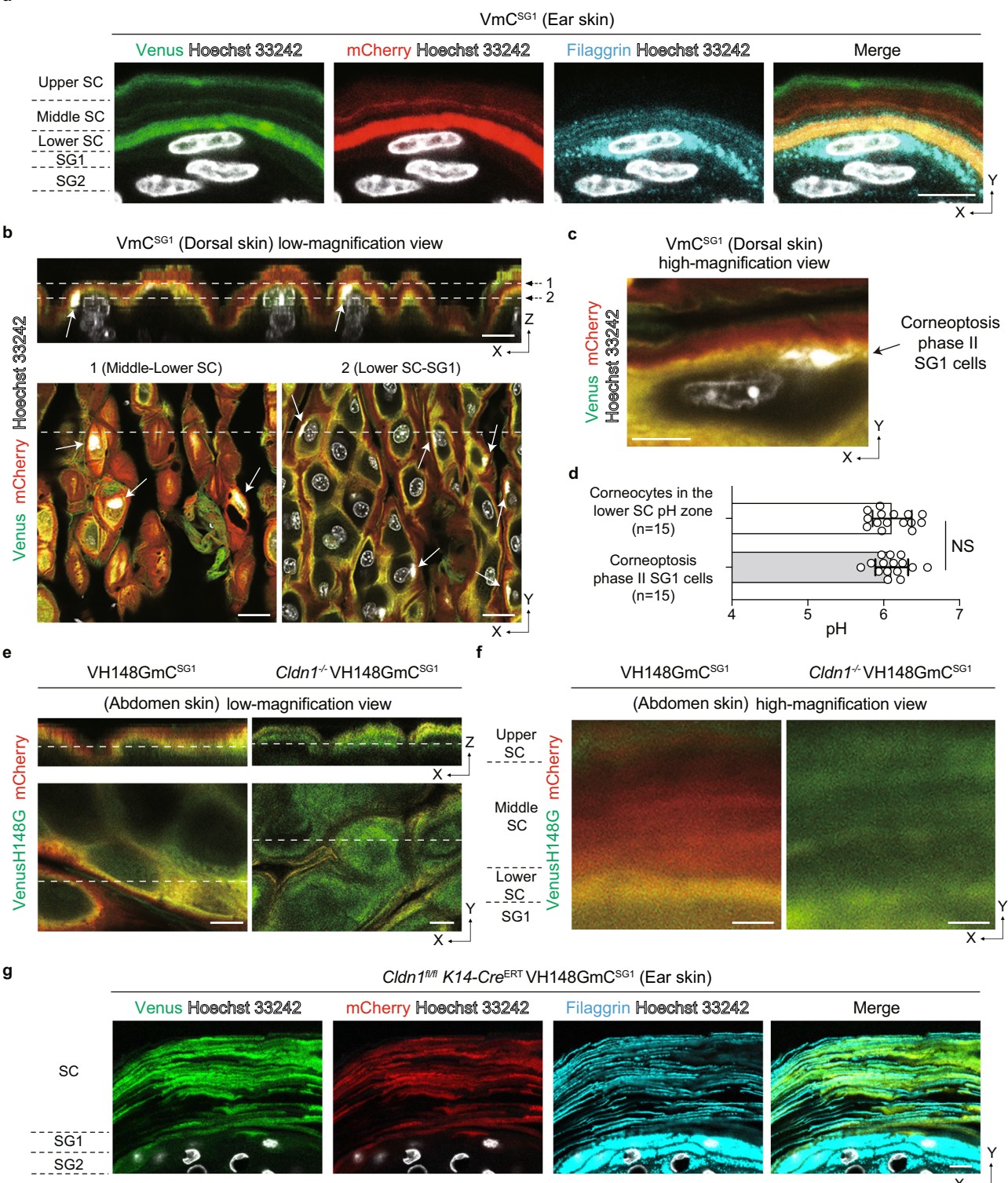

**Fig. 2 | Lower SC-pH zone originates from acidification during corneoptosis, expresses filaggrin, and is hampered by TJ disruption. a** Immunofluorescence microscopy of mouse ear skin from VmC$^{SG1}$ mice visualized for Venus, mCherry, and filaggrin. Scale bar, 5 μm. **b** Z-sliced reconstituted representative (top) and low-magnification X-Y plane (bottom-left and bottom-right) confocal images of the dorsal skin of VmC$^{SG1}$ mice injected with Hoechst 33242 (white). White arrows represent SG1 cells with yellow cytoplasm (corneoptosis phase II SG1 cells). Scale bar, 10 (top) and 20 (bottom-left and bottom-right) μm. **c** High-magnification X-Y plane representative confocal images of the dorsal skin of VmC$^{SG1}$ mice injected with Hoechst 33242 (white). Scale bar, 5 μm. **d** pH values of corneocytes and

SG1 cells in corneoptosis phase II in the lower SC-pH zone ($n = 15$ cells from three biologically independent animals, respectively). **e** Low and **f** high-magnification X-Y plane representative confocal images of the abdominal skin of neonatal VH148GmC$^{SG1}$ and *Cldn1$^{-/-}$* VH148GmC$^{SG1}$ mice. Scale bar, 10 μm (**e**) and 1 μm (**f**). **g** Immunofluorescence microscopy of ear skin from B6.*Cldn1$^{fl/fl}$* K14-CreERT VH148GmC$^{SG1}$ mice to visualize Venus, mCherry, and filaggrin. Scale bar, 10 μm. Data are shown as mean ± SEM and are pooled from three experiments (**d**) or are representative of at least three independent experiments (**a**–**c, e**–**g**). NS, not significant, two-sided Mann–Whitney's test (**d**). Source data are provided as a Source Data file.

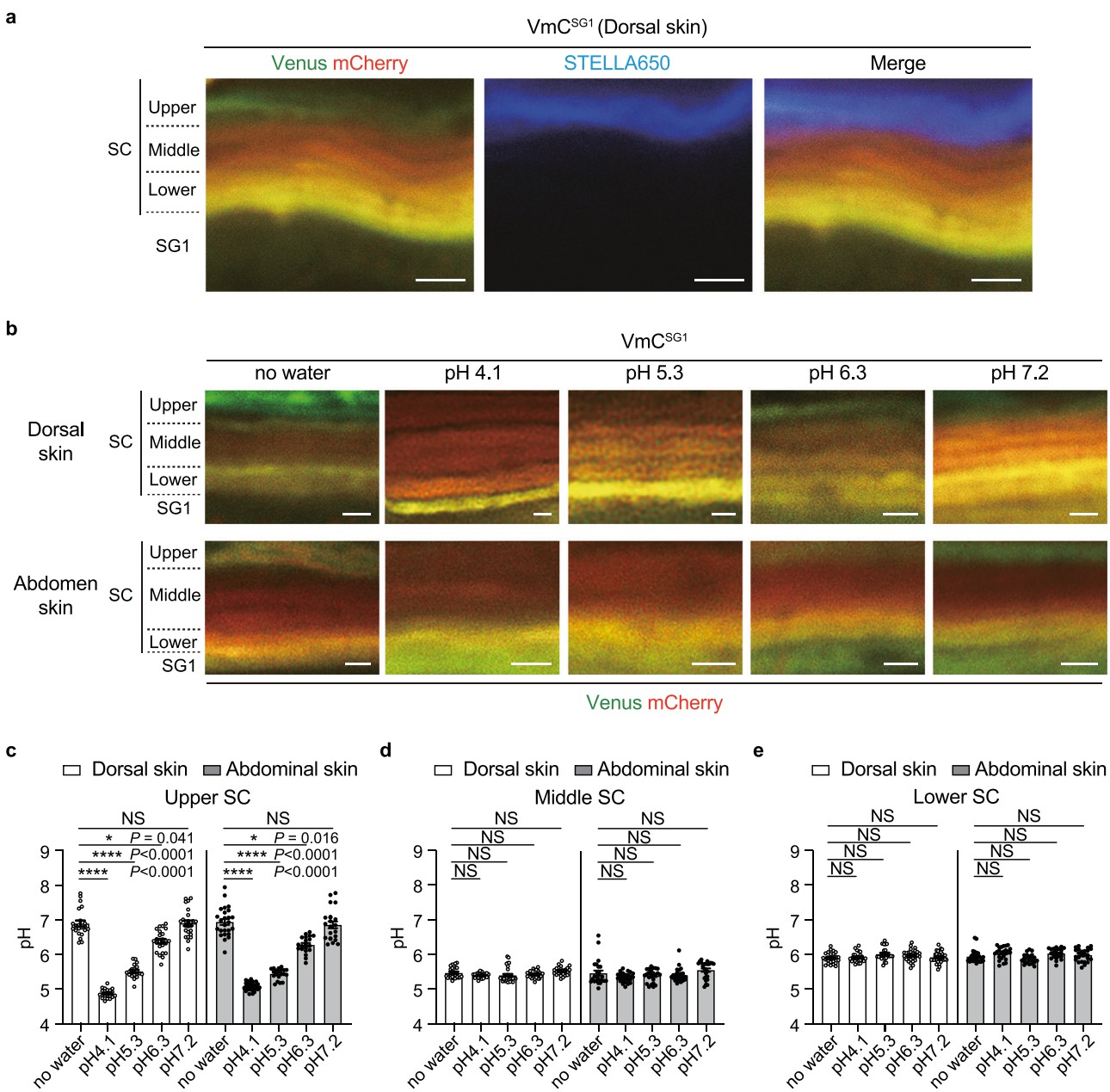

**Fig. 3 | Upper SC-pH zone shows an acclimation-like property. a** High-magnification X-Y plane representative confocal images of the dorsal skin of VmC^SG1 mice soaked in STELLA Fluor 650. Merged images of Venus (green), mCherry (red), and STELLA Fluor 650 (blue) are shown. Scale bar, 1 μm. **b** High-magnification X-Y plane representative confocal images of the dorsal and abdominal skin of VmC^SG1 mice treated with or without various pH buffers (pH 4.1, 5.3, 6.3, and 7.2). Scale bar, 5 μm. **c** pH in the upper SC-pH zones of dorsal or abdomen skin of VmC^SG1 mice treated without buffer (no water) and various pH buffers (pH 4.1, 5.3, 6.3, and 7.2) (*n* = 22, 20, 20, 24, 25, 25, 30, 20, 20, and 22 spots from three biologically independent animals, respectively). **d** pH in the middle SC-pH zones of dorsal or abdomen skin of VmC^SG1 mice treated without buffer (no water) and various pH buffers (pH 4.1, 5.3, 6.3, and 7.2) (*n* = 25, 20, 33, 27, 25, 25, 25, 25, 25, and 25 spots from three biologically independent animals, respectively). **e** pH in the lower SC-pH zones of dorsal or abdomen skin of VmC^SG1 mice treated without buffer (no water) and various pH buffers (pH 4.1, 5.3, 6.3, and 7.2) (*n* = 25, 25, 25, 28, 29, 28, 23, 20, 21, and 25 spots from three biologically independent animals, respectively). Data are shown as mean ± SEM and are pooled from three experiments (**c**–**e**) or are representative of at least three independent experiments (**a** and **b**). *$p < 0.05$, ****$p < 0.0001$; Kruskal-Wallis one-way ANOVA followed by Dunn's post hoc test (**c**–**e**). Source data are provided as a Source Data file.

## pH of the upper SC-pH zone is neutralized by the skin microbiota

As skin microbiota inhabit the SC surface, we assessed whether their presence is required to maintain the pH of the upper SC-pH zone. We treated B6.VmC^SG1 specific-pathogen-free (SPF) mice with an antibody cocktail that has been previously reported to reduce skin microbiota burden and performed intravital SC-pH imaging (Fig. 4a, b, and Supplementary Movie 4). We found that most of the upper SC-pH zone of

mice that received antibiotics became acidic, and the upper-neutral SC-pH zone reduced significantly, indicating that skin microbiota can neutralize the pH of the upper SC-pH zone. To confirm this, we generated VmC^SG1 germ-free C57BL/6 (B6.VmC^SG1 GF) mice and performed intravital SC-pH imaging (Fig. 4c, d, and Supplementary Movie 4). We discovered that the pH of the upper SC-pH zone of B6.VmC^SG1 GF mice (pH 6.05 [95% CI 5.97–6.13]) were significantly lower than that of B6.VmC^SG1 SPF mice (pH 6.65 [95% CI 6.58–6.72]). However, B6.VmC^SG1

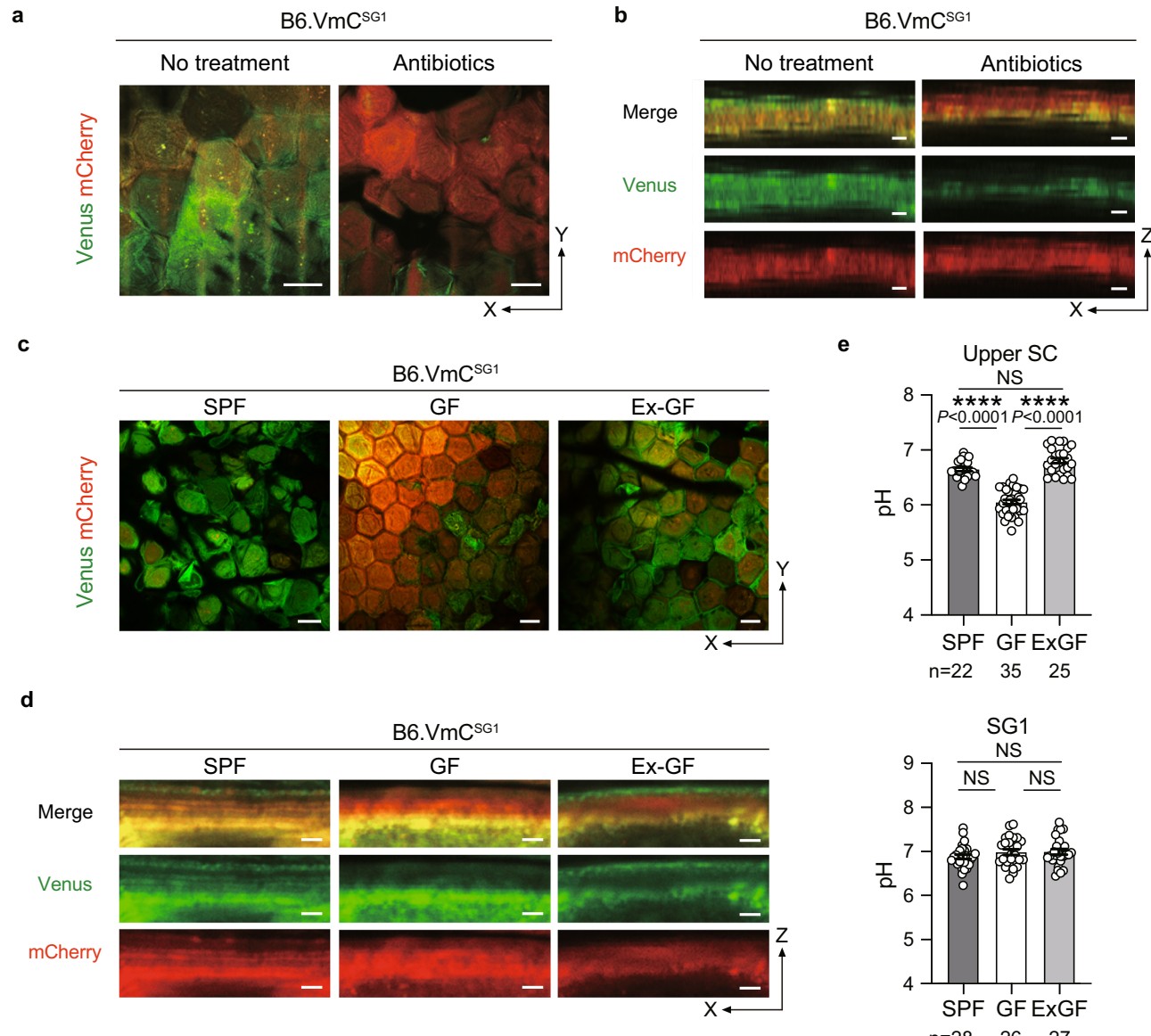

**Fig. 4 | Upper SC-pH has a nearly neutral pH owing to skin microbiota. a** Low-magnification X-Y plane and (**b**) Z-sliced reconstituted representative confocal images of B6.VmC^SG1 mice that received no treatment or antibiotics. Scale bar, 10 μm (**a**) and 1 μm (**b**). **c** Low-magnification X-Y plane and (**d**) Z-sliced reconstituted representative confocal images of SPF, GF, and co-housed B6.VmC^SG1 (Ex-GF) mice. Scale bar, 20 μm (**c**) and 2 μm (**d**). **e** pH in the upper SC of SPF, GF, and co-housed B6.VmC^SG1 mice (top, *n* = 22, 35, and 25 fields from three biologically independent animals, respectively) and SG1 of SPF, GF, and co-housed B6.VmC^SG1 mice (bottom, *n* = 28, 26, and 27 fields from three biologically independent animals, respectively). Data are shown as mean ± SEM and are pooled from three experiments (**e**) or are representative of at least three independent experiments (**a–d**). ****p < 0.0001; NS not significant, Kruskal-Wallis one-way ANOVA followed by Dunn's post hoc test (**e**). Source data are provided as a Source Data file.

GF mice co-housed with B6.VmC^SG1 SPF mice (termed B6.Ex-GF mice) displayed a neutralized upper SC with similar pH values (pH 6.81 [95% CI 6.71–6.90]) to B6.VmC^SG1 SPF mice (Fig. 4e). Collectively, these results suggest that the presence of the skin microbiota neutralizes the upper-nearly neutral SC-pH zone.

## Middle-acidic SC-pH zone serves as a protective barrier against *Staphylococcus aureus*

*S. aureus*, a prominent skin pathogen, is typically associated with inflammatory skin diseases such as AD. The abundance of *S. aureus* is reportedly high on the skin of AD patients, and its distribution differs from that seen in healthy controls[34]. However, how *S. aureus* localizes to the SC in AD remains unclear. To observe the localization of *S. aureus* in the SC, we inoculated the ear skin of B6.VmC^SG1 SPF (hereinafter termed B6.VmC^SG1) mice, ensuring the absence of *S.*

*aureus*, with violet-excitable fluorescent protein (VFP)-labeled *S. aureus* and performed intravital SC-pH imaging (Fig. 5a and Supplementary Fig. 4a). Although *S. aureus* has been reported to localize to the SC surface[35], we found that it also localized at the bottom of the upper-neutral SC pH zone (Fig. 5b and Supplementary Movie 5). Notably, *S. aureus* distribution was restricted until just above the middle-acidic SC-pH zone, and there were no signs of *S. aureus* in the middle-acidic SC-pH zone for at least three weeks (Supplementary Fig. 4b).

Next, we examined the distribution of *S. aureus* upon inflammation. We applied MC903 on the ears of B6.VmC^SG1 mice to induce AD-like dermatitis (Supplementary Fig. 4c). The three stepwise SC-pH zones were maintained; however, all three zones became thicker in mice that received MC903 (likely owing to skin inflammation) when compared to those of vehicle-treated mice (Supplementary Fig. 4d, e). There was no difference in the pH of the upper (vehicle

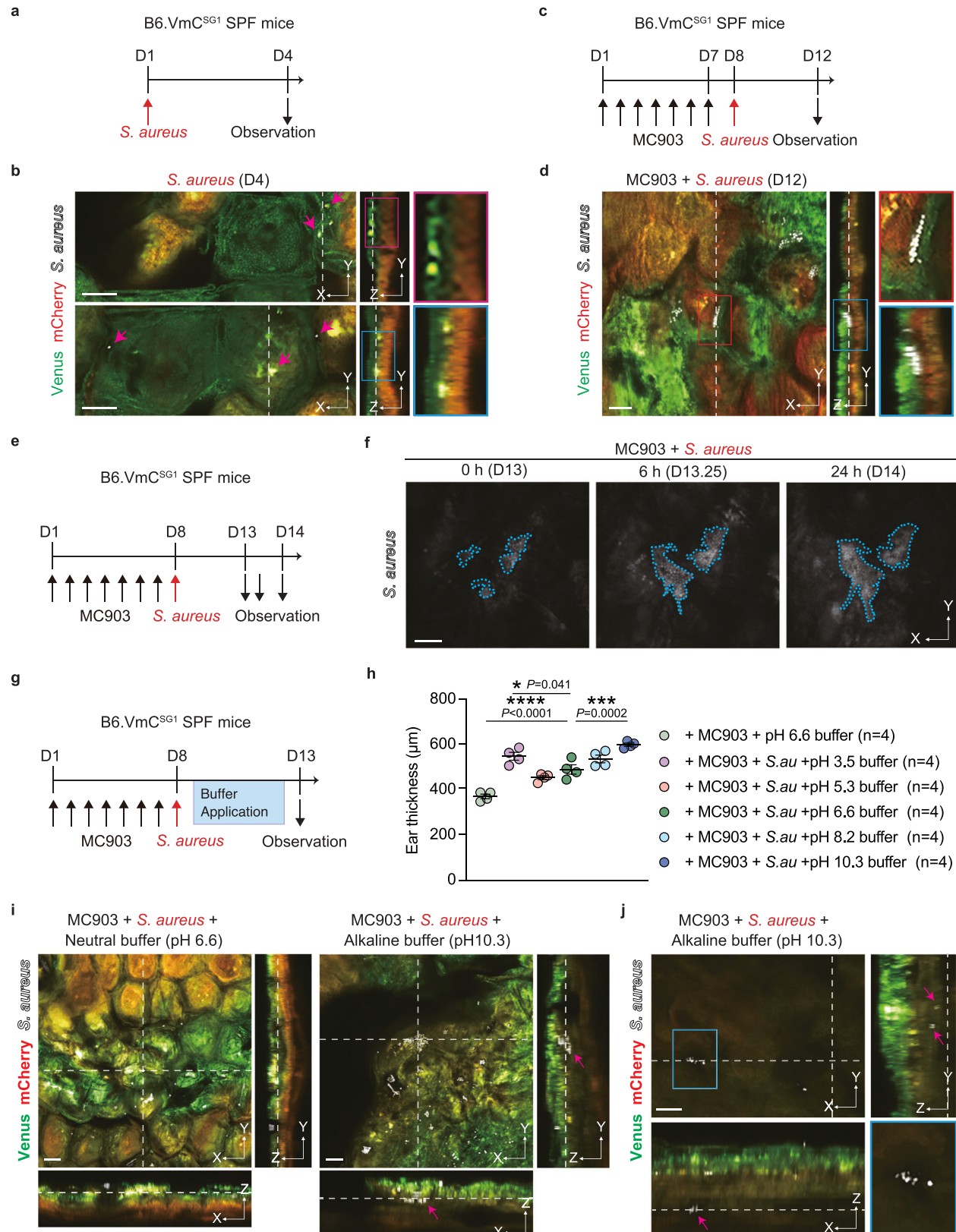

h

Ear thickness (μm)

* P=0.041
**** P<0.0001
*** P=0.0002

- ◯ + MC903 + pH 6.6 buffer (n=4)
- ◯ + MC903 + *S.au* +pH 3.5 buffer (n=4)
- ◯ + MC903 + *S.au* +pH 5.3 buffer (n=4)
- ● + MC903 + *S.au* +pH 6.6 buffer (n=4)
- ◯ + MC903 + *S.au* +pH 8.2 buffer (n=4)
- ◯ + MC903 + *S.au* +pH 10.3 buffer (n=4)

[pH 6.93 ± 0.27] vs MC903 [pH 6.99 ± 0.28]) or lower SC-pH zones (vehicle [pH 6.11 ± 0.12] vs MC903 [pH 6.27 ± 0.16]) between the two groups; however, the middle SC zone tended to have a higher pH in MC903-treated mice than in vehicle-treated mice (vehicle [pH 5.39 ± 0.27] vs MC903 [pH 5.91 ± 0.09]) (Supplementary Fig. 4f).

Subsequently, we applied VFP-labeled *S. aureus* to the ears of B6.VmC^SG1 mice that received MC903 (Fig. 5c, d). We found that *S. aureus* colonized at the bottom of the upper-neutral SC pH zone under MC903-induced dermatitis. Similar to the steady-state control, there were no *S. aureus* signals in the middle-acidic SC-pH zone (Fig. 5d and Supplementary Movie 5). In addition, we performed time-lapse

**Fig. 5 | Middle-acidic SC-pH zone prevents *S. aureus* invasion. a**, **b** B6.VmC^SG1 SPF mice were inoculated with *S. aureus* on day 1. Mice were analyzed on day 4. **a** Experimental design. **b** Low-magnification X-Y plane (left) and low (middle) or high (right, region marked by red and blue lines in the middle) Z-sliced reconstituted representative confocal images. Pink arrows represent *S. aureus* signals. **c**, **d** MC903 was applied to B6.VmC^SG1 SPF mice daily for 7 days (days 1–7) to induce dermatitis (MC903-treated mice). The mice were subsequently inoculated with *S. aureus* on day 8 and analyzed on day 12. **c** Experimental design. **d** Low-magnification X-Y plane (left) and Z-sliced reconstituted (middle) representative confocal images. High-magnification X-Y plane (upper-right, region marked by red lines in left) and Z-sliced reconstituted (lower-right, region marked by blue lines in middle) representative confocal images. **e**, **f** MC903-treated mice were inoculated with *S. aureus* on day 8 and analyzed on days 13, 13.25, and 14. **e** Experimental design. **f** Representative time-lapse images of *S. aureus* detection in the SC of B6.V-

mC^SG1 SPF mice. The blue dashed line represents the area of the *S. aureus* signal(s). **g**–**j** MC903-treated mice were inoculated with *S. aureus* on day 8 and various pH buffers once daily on days 9–12. Mice were analyzed on day 13. **g** Experimental design. **h** Ear thickness of mice which were topically treated with various pH buffers (*n* = 4 each) after MC903 and *S. aureus* application. **i** Low-magnification X-Y plane and Z-sliced reconstituted representative confocal images of mice that received neutral (pH 6.6) and alkaline (pH 10.3) buffers. **j** Representative X-Y plane (upper-left, low-magnification: lower-right, a region marked by blue lines in upper left) and Z-sliced reconstituted representative confocal images of *S. aureus* entered the living cell layer (below SC) in mice that received alkaline (pH 10.3) buffer. The pink arrow represents *S. aureus*. Data are representative of at least three independent experiments (**b**, **d**, **f**, **h**–**j**). Scale bar, 10 μm (**b**, **d**, **f**, **i**, **j**). Data are shown as mean ± SEM (**h**). \*\*\**p* < 0.001; one-way ANOVA with Dunnett's comparison test (**h**). Source data are provided as a Source Data file.

imaging of the SC of B6.VmC^SG1 mice that received MC903 and VFP-labeled *S. aureus*. We found that inoculated *S. aureus* proliferated horizontally at the bottom of the upper-nearly neutral SC-pH zone (Fig. 5f). Furthermore, we found that *S. aureus* proliferation was more suppressed under acidic conditions than under neutral or alkaline conditions in vitro (Supplementary Fig. 4g). Collectively, these findings indicate that the bottom of the upper-nearly neutral SC-pH zone act as a niche for *S. aureus* colonization.

Given that the middle-acidic SC-pH zone showed no signals of *S. aureus*, we next examined the impact of acidic pH in the middle SC-pH zone. After we applied MC903 on the ears of B6.VmC^SG1 mice for 7 days (days 1–7), the mice were applied VFP-labeled *S. aureus* on day 8 and different pH buffers once daily on days 9–12 (four days) to the ears, and analyzed on day 13 (Fig. 5g). The B6.VmC^SG1 mice that received pH 10.3 buffer showed enhanced skin inflammation, thereby exhibiting significantly thicker ear skin than that of mice that received the pH 6.6 buffer (Fig. 5h). Intravital imaging of *S. aureus* and SC-pH distribution revealed that while in mice that received pH 6.6 buffer *S. aureus* colonized horizontally at the bottom of the upper-nearly neutral SC-pH zone, the application of pH 10.3 buffer neutralized the middle SC-pH zones, enabling *S. aureus* to enter into middle or lower SC-pH zones (Fig. 5i) and even into the SG (living cell layer) (Fig. 5j and Supplementary Fig. 4h and Supplementary Movie 6). Thus, these results indicate that the middle-acidic SC-pH zone is the acid mantle layer that serves as a protective barrier against *S. aureus*.

**Three stepwise SC-pH zones provide a logical basis for controlled protease activation**

The SC thickness is maintained by constant shedding of its outermost corneocyte layer via a process known as desquamation[36]. Desquamation is executed via the degradation of corneodesmosomes (involved in corneocyte–corneocyte adherence) by a member of the kallikrein-related peptidase (KLK) family[9,37]. The pH profile of the SC has been considered to regulate the catalytic activity of KLKs and ensure that desquamation occurs at the appropriate location in the SC[2,9]. Under normal conditions, the neutral pH in the lower SC enables lymphoepithelial Kazal-type-related inhibitor (LEKTI) to inactivate KLKs (KLK5, KLK7, and KLK14); however, acidic conditions in the uppermost layer inhibit LEKTI, allowing KLKs to function and promote skin peeling[38,39]. Herein, we revealed a stepwise pH profile with pH values of 7.0, 6.0, 5.4, and 6.7 (pH7.0_6.0_5.4_6.7 model) at the SG1, lower SC, middle SC, and upper SC layers, respectively. To elucidate the biological significance of this stepwise pH profile, we built a mathematical model to understand the pH-dependent regulation of the effective catalytic activity of KLK in SCs with different pH profiles. We modeled the pH-dependent processes of (i) activation from pro-KLKs to active KLKs, (ii) inhibition of active KLKs by LEKTI, (iii) degradation of LEKTI, and (iv) proteolysis by active KLKs (Fig. 6a) (see Methods).

The stepwise SC-pH profile reproduced the increase in effective KLK catalytic activity toward the SC surface, thereby indicating the

necessity of an upper-neutral SC-pH zone at the SC surface for desquamation (Fig. 6b). Compared to the pH7.0_6.0_5.4_6.7 stepwise model, the pH7.0_5.4 gradient model showed decreased effective KLK catalytic activity toward the SC surface (Fig. 6c). This gradient pH profile could not produce the microenvironment required to execute desquamation at the SC surface. In addition, the pH 7.0_5.4_6.7 gradient model demonstrated an earlier rise in KLK activity from the middle SC but a slower KLK activity rise at the upper SC when compared to the pH7.0_6.0_5.4_6.7 stepwise model. This observation suggests that the stepwise pH profile is better suited to modulate the on-off switching of KLK catalytic activity in the upper-neutral SC-pH zone (Fig. 6d).

We also simulated the impact of the effective catalytic activity of KLK on disrupted SC-pH profiles, the uniform pH 5.4 and uniform pH 7.0 models. The uniform pH5.4 model failed to increase the effective catalytic KLK activity throughout the SC region (Supplementary Fig. 6c) because of the low pH that cannot exploit the potential KLK activity compared to the pH7.0_6.0_5.4_6.7 stepwise model and could not initiate desquamation at the SC surface. In contrast, the uniform pH 7.0 model resulted in an abrupt increase in effective KLK catalytic activity in the bottom half of the SC owing to the high pH that fully exploits the potential KLK activity in this region (Supplementary Fig. 6c). The high effective KLK catalytic activity may initiate desquamation even in the bottom half of SC, which leads to abnormal skin phenotype. These results indicate the biological necessity of having several zones of different pH levels in the SC, as for the pH7.0_6.0_5.4_6.7 stepwise model.

Finally, we examined the impact of suppressing KLK catalytic activity in the upper SC-pH zone on desquamation in vivo. This was performed by applying a pH 3.5 buffer, that significantly suppresses the catalytic activity of mouse KLK compared with a pH 6.6 buffer, on the ear of B6.VmC^SG1 mice for four consecutive days (Supplementary Fig. 6d, e). The mice that received the pH 3.5 buffer maintained three stepwise SC-pH zones; however, they showed thicker SC than the mice that received the pH 6.6 buffer (Supplementary Fig. 6e). These results suggest that suppressing KLK catalytic activity leads to the suppression of desquamation, thereby promoting SC thickening. These results were compatible with our mathematical model.

Overall, the simulation results suggest that the three stepwise pH zones trigger LEKTI to associate with KLKs to suppress their catalytic activity in the lower-moderately acidic SC-pH zone. This also allows KLKs to remain in an inactive state in the middle-acidic SC-pH zone and become activated in the upper-nearly neutral SC-pH zone to execute desquamation. Furthermore, our results suggest that keratinocytes undergo differentiation, thereby developing three-tiered pH zonation, which maintains the thickness and barrier function of the SC to produce SC homeostasis (Fig. 6e).

## Discussion

While past studies examined SC-pH using tools that are largely restricted to measuring extracellular pH in the SC (such as applying a

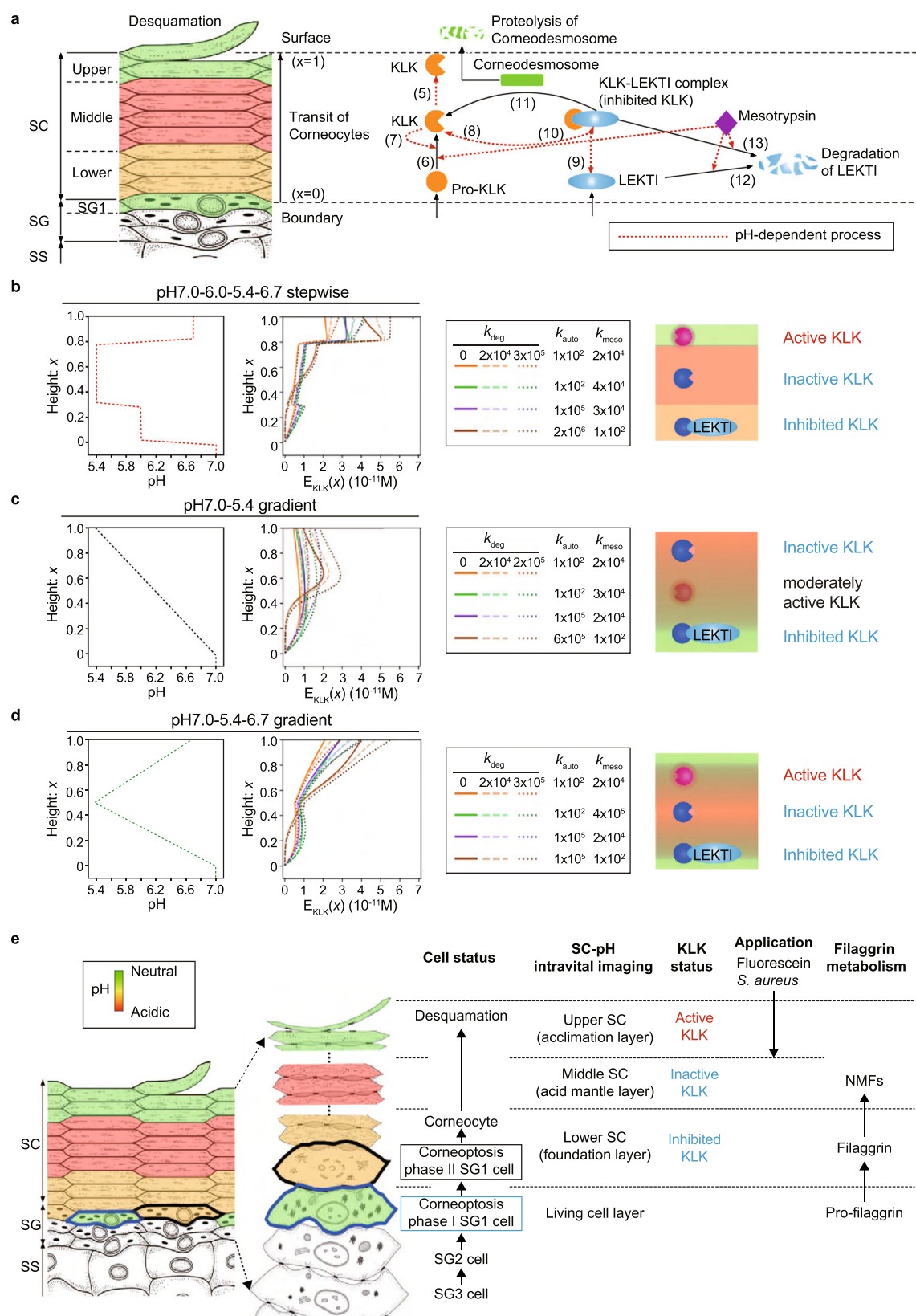

pH electrode to the skin), we conducted through high-resolution intravital imaging of intracellular pH in the SC using VmC^SG1 and VH148GmC^SG1 mice. Our findings showed that the SC has three distinct stepwise pH zones (lower-moderately acidic [pH 6.0], middle-acidic [pH 5.4], and upper-nearly neutral [pH 6.7]). The three SC-pH zones were commonly found across various body parts, implying

that they play a role in SC homeostasis. Furthermore, the extra-cellular pH in the SC of corneocytes using B6.Display-VmC^SG1 mice exhibited a three-tiered zonation, and the pH value matched the intracellular pH. A possible explanation for the discrepancy of our data from previous studies may be that the previous study used methods that induce skin inflammation or dissolve intercellular

**Fig. 6 | Three stepwise pH zones provide a favorable environment for SC desquamation. a** Overview of pH-dependent processes included in a mathematical model of catalytic KLK activity in the SC. The corneocytes mobilize from the SG/SC boundary toward the SC surface via epidermal turnover. Inactive pro-KLKs and LEKTI are released into the SC/SG boundary from SG1 cells. Subsequently, pro-KLKs are activated into KLKs (active form) by KLKs and mesotrypsin. KLKs are capable of degrading corneodesmosomes via proteolysis, thereby initiating desquamation. LEKTI inhibits the catalytic activity of KLKs by reversibly forming a KLK−LEKTI complex, which inactivates KLK. Mesotrypsin degrades LEKTI in the KLK−LEKTI complex. The numbers in the figure correspond to the model equations. The red dashed line represents the pH-dependent process. **b**–**d** Profiles of effective catalytic KLK activity ($E_{KLK}(x)$), parameters, and schematic of (**b**) KLK and LEKTI in the pH7.0_6.0_5.4_6.7 stepwise model, (**c**) pH 7.0_5.4 gradient model, and (**d**) pH7.0_5.4_6.7 gradient model. **e**. Schematic and summary of the function of three stepwise SC-pH zones. The image of SC and SG (right) in Fig. 6e was adapted with permission from Oxford University Press: International Immunology "Dissecting the formation, structure, and barrier function of the stratum corneum" by Matsui T et al. Copyright 2015.

lipids in the SC[12–14], both of which can affect SC-pH, whereas our intravital imaging was non-invasive.

Various proteases localize throughout the SC[40]; however, we confirmed that both Venus-mCherry/VenusH148G-mCherry proteins remained abundantly intact in the SC layer. Based on this evidence and the acclimation-like property of the upper SC, we successfully measured the fluorescence changes in Venus and mCherry by applying the different pH buffers to the upper SC zones. These results suggest that the expression of ratiometric pH probe proteins in SG1 cells and their integration into the corneocytes are useful methods to measure and visualize SC-pH.

The SC comprises layers of corneocytes with intercellular spaces sealed by lipid lamellar structures (the "brick and mortar model"). The insolubility of the SC has hampered comprehensive analysis, and the SC has been considered a homogeneous barrier[27]. However, using immune-electron microscopy of filaggrin, SC can be divided into lower SC with filaggrin and upper SC without filaggrin[26]. Furthermore, scanning electron microscopy of the SC demonstrated that the lower SC swells when soaked in water, forming massive water inclusions between adjacent cell layers. In contrast, the middle SC remains nearly unaffected by water stress and maintains its thickness. The upper SC also swells massively, thereby loosening its intracellular filament packing. These results suggested that the SC can be divided into three distinct regions based on their water penetration and binding potentials[41]. In addition, our previous study conducted using TOF-SIMS, which enables the visualization of the distribution of natural substances without using staining methods, revealed that arginine, a major component of filaggrin-derived natural moisturizing factors, was concentrated in the middle SC, suggesting that this layer plays a key role in skin hydration[28]. The soaking assay in the study revealed that the upper SC allowed for the influx and efflux of various metal ions, while the middle and lower layers exhibited distinct barrier properties. Through immunofluorescence staining of filaggrin and intravital SC-pH imaging using STELLA 650 containing Milli-Q water, we observed filaggrin and STELLA 650 fluorescence located in the lower SC-pH zone and the upper SC-pH zone, respectively. These results suggest that the three zones observed by TOF-SIMS correspond to the three SC-pH zones. Thus, the SC is not a simple monotonous layer but rather differentiates into three sharply demarcated layers, with pH being a key regulator of this process.

It is unclear how pH regulation is altered in the SC, resulting in the development of three stepwise SC-pH zones. A recent proteome study showed that the peptides of vacuolar-type ATPase (which pumps protons across the plasma membranes using ATP hydrolysis energy) are detected from SC samples[42]. However, mitochondria do not exist in the corneocytes, and ATP synthesis by oxidative phosphorylation does not occur in the SC. Thus, it is unclear whether vacuolar-type ATPase is active in the SC and plays a role in maintaining SC-pH. Cryo-electron microscopy (cryo-EM) revealed that the organization of the lipid lamellae drastically changes in the intercellular spaces between the first and the fifth cells of the SC: (1) a single-band pattern with 2.0– 2.5 nm periodicity, and (2) a two-band pattern with 5.5–6.0 nm periodicity[43]. Since the optimal pH of the lipid metabolism enzyme is acidic, we speculate that the lipid metabolism enzymes may contribute to forming lipid lamellae organization thereby regulating the SC-pH.

Given that the upper SC-pH zone has acclimation-like capability and passively adjusts its pH to the external environment and that skin microbiota localizes on the SC surface, we speculated that the skin microbiota affects the pH of the upper SC-pH zone. Indeed, we found that GF mice had a slightly more acidic upper SC-pH zone than SPF mice, suggesting that skin microbiota contributes to neutralizing the upper SC-pH zone. Notably, *S. aureus* localized in steady-state conditions and colonized the bottom surface of the upper SC-pH zone in inflammatory conditions. In vitro *S. aureus* proliferation assay results revealed that *S. aureus* favors a neutral microenvironment over acidic conditions. Similar to *S. aureus*, most bacteria thrive at neutral pH levels (optimally at pH 7.0), and deviation from these optimal pH values can reduce their growth rate by approximately 50%[21,44]. Thus, the bottom surface of the upper SC-pH zone may act as a proliferation niche for pathogens. Furthermore, we found that *S. aureus* could still proliferate at pH 5.4, the pH value of the middle-acidic SC-pH zone, even though there were no *S. aureus* signals. Based on these results, we speculate that the middle-acidic SC-pH zone can not only suppress *S. aureus* proliferation, but also maintains structural integrity and/or antimicrobial peptide production to achieve its barrier function.

The SC is unique as it can maintain homeostasis and differentiate into three stepwise SC-pH zones, although it consists of dead keratinocytes. Inflammatory skin diseases, such as psoriasis, lead to the development of nucleated keratinocytes in the SC (called parakeratosis) and a thicker SC (called acanthosis) when compared to normal healthy skin, suggesting abnormal SC homeostasis. In addition, patients with ichthyoses (a group of heterogenous genetic skin diseases) exhibit a thickened and abnormally desquamated SC. Some individuals show the upregulation of epidermal differentiation-related genes and downregulation of TJ component-associated genes, implying impaired SC differentiation[45]. Thus, deciphering the mechanisms underlying stepwise SC-pH zones generation and SC-pH regulation may provide novel insights for developing treatment strategies for inflammatory skin diseases and ichthyoses.

## Methods
### Mice
C57BL/6 (B6) and hairless (HR) mice were obtained from Japan SLC Inc. (Hamamatsu, Japan). B6.*Cldn1*[+/-] mice were provided by Dr. M. Furuse (National Institute for Physiological Sciences, Okazaki, Japan). B6.*SASP*[EGFP/+] (B6.EGFP[SG1]) mice were generated by integrating the EGFP-Neo cassette into the SASP gene locus on chromosome 6 in B6 background ES cells via homologous recombination[10]. To generate B6.*SASP*[VenusH148G-mCherry/+] (B6.VH148GmC[SG1]) mice, we modified the knock-in vector of B6.*SASP*[EGFP/+] mice by replacing the EGFP-Neo cDNA with Venus[H148G]-mCherry cDNA using pSK-VenusH148G-mCherry[10] and CRISPR/Cas9. Furthermore, the Venus[H148G] mutation was reverted to wild-type (G148 to V148) using oligonucleotide-mutagenesis via CRISPR/Cas9 to generate B6.*SASP*[Venus-mCherry/+] (B6.VmC[SG1]) mice (Supplementary Fig. 1a). To generate B6.SASP[Display-Venus-mCherry/+] (B6.Display-VmC[SG1]) mice, a pDisplay-Venus-mCherry cassette was integrated into the SASP gene locus on chromosome 6 in B6 background ES cells via homologous recombination. Venus-mCherry fusion protein expressed from the pDisplay Mammalian Expression Vector (Invitrogen) was

fused at the N-terminus to the murine Ig κ-chain leader sequence, which directs the protein to the secretory pathway, and at the C-terminus to the PDGFR transmembrane domain, which anchors the protein to the plasma membrane, displaying it on the extracellular side. B6.EGFP[SG1], B6.VH148GmC[SG1], B6.VmC[SG1], and B6.Cldn1[+/-] mice were backcrossed with HR mice for at least 10 generations to produce EGFP[SG1], VH148GmC[SG1], VmC[SG1], and Cldn1[+/-] mice, respectively. Cldn1[+/-] mice were intercrossed with VH148G mCherry[SG1] mice to generate Cldn1[-/-] VH148GmCherry[SG1] mice. Epidermis-specific Cldn-1 conditional knockout mice (B6.Cldn1[fl/fl] K14-CreERT) were obtained from RIKEN LARGE (Accession No. CDB0803K)[46] and intercrossed with VH148GmC[SG1] mice to produce B6.Cldn1[fl/fl] K14-CreERT VH148GmC[SG1] mice. For epidermis-specific Cldn1 knockout induction, B6.Cldn1[fl/fl] K14-CreERT VH148GmC[SG1] mice received intraperitoneal injection of 1 mg tamoxifen for 5 days and the mice were used for the experiment nine days after the last tamoxifen injection. To generate B6.VmC[SG1] GF mice, germ-free rederivation of B6.VmC[SG1] mice was performed at the gnotobiotic animal facilities of Keio University. B6.VmC[SG1] GF mice were kept in individual gnotobiotic isolators until analysis with 12 h light/12 h dark cycle at a controlled temperature (23 ± 2 °C) and humidity (50% ± 10%) with free access to autoclaved water and autoclaved standard laboratory chow diet. For co-housing, SPF and GF mice were housed in the same cage under SPF conditions for seven days. All mice except B6.VmC[SG1] GF mice were maintained and housed at pathogen-free animal facilities at the RIKEN Yokohama Branch or Keio University School of Medicine with 12 h light/12 h dark cycle at a controlled temperature (23 ± 2 °C) and humidity (50% ± 10%) with free access to water and standard laboratory chow diet (housed by strains). Sex was not considered in this study. 3–6-month-old mice (except Cldn1[-/-] VH148GmCherry[SG1] mice, which die within one day of birth) were used for all experiments because the characteristics of corneoptosis do not change in this range[10].

The information of the mice used in each figure is as below.

Figure 1: In Figs. 1b, 1 male, and 2 female VmC[SG1] mice at the age of 16–27 weeks were analyzed. In Figs. 1c, d, 2 male and 1 female VmC[SG1] mice at the age of 12–23 weeks were analyzed. In Figs. 1e–g, 3 male and 1 female VmC[SG1] mice at the age of 23–27 weeks were analyzed. In Figs. 1h and i, 2 male and 1 female B6.Display-VmC[SG1] mice at the age of 20–27 weeks were analyzed.

Figure 2: In Figs. 2a, 2 male and 1 female VmC[SG1] mice at the age of 16–21 weeks were analyzed. In Figs. 2b–d, 2 male and 1 female VmC[SG1] mice at the age of 12–15 weeks were analyzed. In Figs. 2e, f, 2 male and 1 female VH148GmC[SG1] mice at the age of 19–22 weeks and 2 male and 1 female neonatal (the age of day 1) Cldn1[-/-] VH148GmCherry[SG1] mice were analyzed. In Figs. 2g, 1 male and 2 female B6.Cldn1[fl/fl] K14-CreERT mice at the age of 12–13 weeks were analyzed.

Figure 3: In Fig. 3a, 3 male VmC[SG1] mice at the age of 13–21 weeks were analyzed. In Figs. 3b–e, 2 male and 1 female VmC[SG1] mice at the age of 12–16 weeks were analyzed.

Figure 4: In Fig. 4a, b, 1 male and 2 female B6.VmC[SG1] mice at the age of 16–23 weeks were analyzed. In Figs. 4c–e, 3 male B6.VmC[SG1] SPF, 3 male B6.VmC[SG1] GF, and 3 male B6.VmC[SG1] Ex-GF mice at the age of 20–23 weeks were analyzed.

Figure 5: In Figs. 5b, d, and f, 2 male and 1 female B6.VmC[SG1] mice at the age of 12-19 weeks were analyzed. In Fig. 5h, 24 male B6.VmC[SG1] SPF mice (4 male B6.VmC[SG1] SPF mice in each group) at the age of 20–23 weeks were analyzed. In Figs. 5i, 6 male B6.VmC[SG1] SPF mice (3 male B6.VmC[SG1] SPF mice in both group) at the age of 20–23 week were analyzed. In Figs. 5j and 3 male B6.VmC[SG1] SPF mice at the age of 20–23 weeks were analyzed.

supplementary Fig. 1: In Supplementary Fig. 1b, 1d, and 1 f, SG1 cells derived from 2 male and 1 female VmC[SG1] mice at the age of 18–22 weeks were analyzed. In Supplementary Fig. 1c, e, and 1g, SG1 cells derived from 3 male VH148GmC[SG1] mice at the age of 12–14 weeks were analyzed.

supplementary Fig. 2: In Supplementary Fig. 2a–c, 2 male and 1 female VH148GmC[SG1] mice at the age of 12–22 weeks were analyzed. In Figs. 2d and e, 1 male and 2 female VmC[SG1] mice at the age of 14–23 weeks were analyzed.

supplementary Fig. 3: In Supplementary Fig. 3a and b, 2 male and 1 female at the age of 12–20 weeks EGFP[SG1] mice and 1 male and 2 female VmC[SG1] mice at the age of 14-23 weeks were analyzed. In Supplementary Fig. 3c, 3 male VH148GmC[SG1] mice at the age of 12–20 weeks were analyzed. In Supplementary Fig. 3d and e, 1 male and 2 female B6.Cldn1[fl/fl] K14-CreERT mice at the age of 12–13 weeks and 2 male and 1 female VmC[SG1] mice at the age of 13–16 weeks were analyzed. In Supplementary Fig. 3f, 1 male and 2 female B6.Cldn1[fl/fl] K14-CreERT mice at the age of 12–13 weeks were analyzed.

supplementary Fig. 4: In Supplementary Fig. 4b, 2 male and 1 female B6.VmC[SG1] mice at the age of 12–19 weeks were analyzed. In Supplementary Fig. 4d-f, 4 male and 2 female B6.VmC[SG1] mice (2 male and 1 female B6.VmC[SG1] SPF mice in both group) at the age of 12–19 weeks were analyzed.

supplementary Fig. 6: In Supplementary Fig. 6e and f, 2 male and 1 female B6.VmC[SG1] mice at the age of 20–23 weeks were analyzed.

## pH calibration using isolated SG1 cells

SG1 cells were isolated from the dorsal skin of VmC[SG1] and VH148GmC[SG1] mice, as previously described, with some modifications[10]. Briefly, mice were anesthetized via intraperitoneal injection of a medetomidine-midazolam-butorphanol (MMB) mixture (0.75:4:5). Subsequently, 9 μg/mL recombinant exfoliative toxin-A (ETA) dissolved in phosphate-buffered saline (PBS) containing 1 mM CaCl2 was injected into the dorsal skin of VmC[SG1] and VH148GmC[SG1] mice and mice placed at 37 °C for 60 min. The ETA sheet, which composed of SG1/2/SC, was then mechanically peeled from the skin. After 5 min treatment of trypsin, Venus-mCherry-positive SG1 cells and Venus-mCherry-negative SG2 cells derived from VmC[SG1] mouse ETA sheet or VenusH148G-mCherry-positive SG1 cells and VenusH148G-mCherry-negative SG2 cells derived from VH148GmC[SG1] mouse ETA sheet were placed into buffers of various pH values (pH 4.1–8.0), with nigericin (10 μM) and valinomycin (10 μM) (both FUJIFILM Wako Pure Chemical Co., Osaka, Japan). The buffers consisted of 50 mM pH buffer and 150 mM NaCl. Acetate and phosphate buffers were used to adjust the pH values to 4.1–5.7 and 5.8–8.0, respectively. The fluorescence intensity of the Venus, VenusH148G, and mCherry proteins in SG1 cells was determined using an HC PL APO CS2 63x/NA 1.40 immersion objective lens on an inverted TCS SP8 confocal microscope (Leica Microsystems, Wetzlar, Germany). A 495/545-nm emission filter was used to detect Venus or VenusH148G fluorescence upon excitation with a 488-nm OPSL laser, and a 580/720-nm emission filter was used for mCherry fluorescence upon excitation with a 552-nm OPSL laser. For image acquisition, 184.88 μm ×184.88-μm planes were scanned at a resolution of 361 nm per pixel with a speed of 400 Hz using a laser with its power set at (i) 3.70 μW for Venus and 2.88 μW for mCherry or (ii) 3.42 μW for Venus and 2.53 μW for mCherry for SG1 cells from VmC[SG1] mice and (iii) 4.93 μW for VenusH148G and 1.73 μW for mCherry or (iv) 3.42 μW for Venus and 1.73 μW for mCherry for SG1 cells from VH148GmC[SG1] mice. X-Y plane images (30) with 0.50-μm Z spacing were constructed for each X-Y plane.

Images were analyzed using Fiji [ImageJ, National Institutes of Health (NIH), Bethesda, MD, USA]. SG1 cell images were projected using the "MAX intensity" script. Thirty SG1 cells were marked by the region of interest (ROI), and their average intensities were measured using the ROI manager. The background intensity was measured similarly. Changes in fluorescence intensity were calculated as the average intensity of the ROI minus the background intensity of each image. The Venus/mCherry and Venus[H148G]/mCherry fluorescence ratios and relative ratios at various pH values were fitted to a sigmoidal nonlinear regression model using GraphPad Prism 9.

## Intravital SC-pH imaging (X-Y- and Z-slice imaging)

Intravital imaging of mice was performed as previously described, with some modifications[10]. Briefly, mice were anesthetized via intraperitoneal injection of a medetomidine-midazolam-butorphanol (MMB) mixture (0.75:4:5). Dorsal, abdomen, ear, tail, and palm skin samples from the mice were attached to a micro cover glass (40 mm × 50 mm thickness 0.17 ± 0.005 mm, C4050HT; MATSUNAMI Glass IND., Osaka, Japan) containing an aliquot of Milli-Q water, Milli-Q water with 10 μM STELLAFluor 650 free COOH (Goryo Chemical, Sapporo, Japan), or buffers of various pH values (4.1–8.0). They were then fixed on a heated stage (TPi-SQH26; Tokai Hit, Fujinomiya, Japan) at 37 °C. Images were acquired using HC PL APO CS2 63x/NA 1.40 or HC PL APO CS2 100x/NA 1.40 using immersion objective lens with an inverted TCS SP8 confocal microscope (Leica Microsystems). A 410/480-nm emission filter was used to detect Hoechst 33342 signals upon excitation using a 405-nm diode laser. A 495/545-nm emission filter was used for Venus and VenusH148G detection upon excitation using a 488-nm OPSL laser, and a 580/720-nm emission filter was used for mCherry detection upon excitation using a 552-nm OPSL laser. For low-magnification image acquisition, 184.7 × 184.7-μm planes were scanned at a resolution of 91 nm per pixel, with the Z spacing between slices being 0.3–0.5 μm. For high-magnification image acquisition, 46.2 × 46.2-μm single X-Y planes were scanned at a resolution of 45 nm per pixel. The laser power was set at (i) 3.70 μW for Venus and 2.88 μW for mCherry (abdominal, toe, and skin) or (ii) 3.42 μW for Venus and 2.53 μW for mCherry (dorsal and ear skin) for image acquisition of VmC$^{SG1}$ mice, (iii) 4.93 μW for VenusH148G and 1.73 μW for mCherry for image acquisition of VH148GmC$^{SG1}$, Cldn1$^{-/-}$ VH148GmCherry$^{SG1}$, and B6.Cldn1$^{fl/fl}$ K14-CreERT VmC$^{SG1}$ mice, and (iv) 64.0 μW for Venus and 54.0 μW for mCherry for image acquisition of B6.Display-VmC$^{SG1}$ mice. In some experiments, 100 μL of saline solution containing 100 μg/mL Hoechst 33342 in distilled water was injected intradermally into the dorsal skin of VmC$^{SG1}$ mice 24 h prior to imaging.

To detect fluorescently labeled *S. aureus* in the SC of murine ear pinnae, B6.VmC$^{SG1}$ mice were inoculated with the bacteria and then anesthetized on a heater pad via intraperitoneal injection of an MMB mixture (0.75:4:5) or a combination of intraperitoneal pentobarbital (10 mg/kg) injection and 0.5–1.5% isoflurane inhalation. The mice were placed in a sealed acrylonitrile-styrene resin box with air vent valves (for inhalation anesthesia) and an observation window covered by a micro cover glass (C4050HT). The ear skin was attached to the micro cover glass. The fluorescence of both the bacteria and the mice was visualized through an FV3000 laser scanning confocal microscope (Olympus) equipped with a UPLSAPO 60XS2 objective lens and the following settings: (1) mAmetrine: excitation, 405 nm; collection, 500–600 nm; (2) Venus: excitation, 514 nm; collection, 530–580 nm; and (3) mCherry: excitation, 594 nm; collection, 610–710 nm. For image acquisition, 212.13 μm × 212.13 μm planes were scanned at a resolution of 0.207 μm per pixel, and the Z spacing between slices was 0.5–1.0 μm. Images were processed using Fiji (ImageJ, NIH) and Photoshop 2023 (Adobe, Mountain View, CA, USA).

## SC-pH measurements

Since the Venus/mCherry fluorescence ratio was normalized to its corresponding value at pH 8.0 (relative Venus/mCherry), the ratiometric curves of the relative Venus/mCherry fluorescence measured under two laser powers overlapped. Dorsal, abdomen, and ear skin samples from VmC$^{SG1}$ mice were attached to a micro cover glass (C4050HT) containing an aliquot of Milli-Q water or pH 8.0 buffer. Subsequently, Venus and mCherry protein fluorescence intensity was measured using an HC PL APO CS2 63x/NA 1.40 oil immersion objective lens on an inverted TCS SP8 confocal microscope (Leica Microsystems GmbH). Similar to the imaging of isolated SG1 cells, 184.88 μm ×184.88-μm planes were scanned at a resolution of 361 nm per pixel with a speed of 400 Hz. The laser power was set at (i) 3.70 μW for

Venus and 2.88 μW for mCherry (abdominal skin) or (ii) 3.42 μW for Venus and 2.53 μW for mCherry (dorsal and ear skin) for image acquisition of VmC$^{SG1}$ or (iii) 64.0 μW for Venus and 54.0 μW for mCherry for image acquisition of B6.Display-VmC$^{SG1}$ mice. Single X-Y plane images were acquired. Images were analyzed using Fiji and Cornecoytes or SG1 cells were marked by the ROI, and their average intensities were measured using the ROI manager. Since the upper SC-pH zone is capable of acclimatization, the Venus/mCherry fluorescence ratio was normalized to the Venus/mCherry ratio of the upper SC-pH zone at pH 8.0 to obtain relative Venus/mCherry values. The SC-pH was calculated by substituting the relative Venus/mCherry values for the ratiometric curves established using isolated SG1 cells.

## Electron microscopic analysis of the epidermis

The dorsal skins of VmC$^{SG1}$ mice were fixed in 2.5% glutaraldehyde in 0.1 M phosphate-buffered saline (PBS) for 2 h. The specimens were washed overnight at 4 °C in the same buffer and post-fixed with 2% osmium tetroxide in 0.1 M phosphate buffer for 2 h at 4 °C. The specimens were dehydrated in a graded ethanol series and embedded in TAAB EPON 812 (Nisshin-EM, Tokyo, Japan), and 0.2-μm sections were cut horizontally. Images were obtained using a SU8220 field emission scanning electron microscope (Hitachi High Technologies Corporation, Tokyo, Japan) with a YAG-backscattered electron detector (5 kV).

## Immunofluorescence staining

Mouse skin samples were directly embedded in optimal cutting temperature compound (Sakura Finetek, Tokyo, Japan) and frozen at −80 °C. Kawamoto's adhesive film (Cryofilm type 2 C (9); SECTION-LAB Co. Ltd., Yokohama, Japan) was applied on the exposed cutting surface, and frozen samples were cut into 5-μm-thick sections[47]. Without drying or fixation, the sections were blocked with 0.1% bovine serum albumin (Sigma Aldrich, St. Louis, MO, USA) in PBS at 25 °C for 30 min, incubated with an anti-filaggrin antibody (1:1000; Poly19058; BioLegend, San Diego, CA, USA) in blocking solution at 25 °C for 1 h or at 4 °C overnight. The sections were washed twice with PBS, incubated with donkey anti-rabbit IgG highly cross-adsorbed secondary antibody, Alexa Fluor™ 647 (1:1000; Thermo Fisher Scientific, Waltham, MA, USA) in blocking solution at 25 °C for 30 min, and washed twice with PBS. The sections were mounted in Dako Fluorescence Mounting Medium (Agilent Technologies, Santa Clara, CA, USA) and observed under an inverted TCS SP8 confocal microscope (Leica Microsystems). Images were processed using Fiji (ImageJ, NIH) and Adobe Photoshop (Adobe).

## Western blot analysis of murine SC

The SC layer from the dorsal skin of EGFP$^{SG1}$, VmC$^{SG1}$, and B6.Cldn1$^{fl/fl}$ K14-CreERT VH148GmC$^{SG1}$ mice was obtained via tape stripping (TS) with the use of large D-Squame tape strips (Promotool, Inc., Chatou, France). TS was performed nine (for EGFP$^{SG1}$ and VmC$^{SG1}$ mice) or seven (for B6.Cldn1$^{fl/fl}$ K14-CreERT VH148GmC$^{SG1}$ mice) times, and each time, the tape was pressed against the skin for 5 s with standardized pressure (225 gr/cm$^2$) using a D-Squame pressurizer (Promotool, Inc.). Each tape was soaked in 1.8 mL of urea buffer (5 M urea/2 M thiourea), PBS, and 1 mM EDTA supplemented with a protease inhibitor cocktail (Nacalai Tesque, Tokyo, Japan) for 5 min. The lysates were centrifuged at 15,000 × *g* for 10 min at 4 °C, and the protein concentrations of the supernatants were estimated using the SensoLyte OPA Protein Quantification Kit Fluorimetric (AnaSpec Inc., Fremont, CA, USA). The supernatants were denatured at 95 °C for 5 min in Laemmli buffer (Bio-Rad, Hercules, CA, USA) and the denatured samples were separated using Tris/glycine/SDS buffer. Following electrophoresis, proteins were transferred onto a PVDF membrane (Bio-Rad). Membranes were incubated for 5 min at room temperature (25 °C) in EveryBlot blocking buffer and then with a 1:1000 dilution of anti-GFP (598; MBL Life Science, Sunnyvale, CA, USA), anti-RFP (PM005; MBL Life Science), 1:1000

dilution of anti-filaggrin (Poly19058; BioLegend), or 1:100 dilution of anti-cytokeratin 10 (LH2; Santa Cruz Biotechnology, Dallas, TX, USA) primary antibody at 4 °C. Immune complexes were detected using anti-rabbit or anti-mouse IgG, horseradish peroxidase-linked secondary antibody, Clarity Western ECL Substrate (all Bio-Rad), and an iBright FL1500 Imaging system (Thermo Fisher Scientific).

## Development of violet-excitable fluorescent protein-labeled *Staphylococcus aureus*

mAmetrine-expressing MW2 *S. aureus* was generated via allele replacement using the pRN112 plasmid as previously described[48]. Briefly, pRN112 was introduced into MW2 *S. aureus* by electroporation, and the cells were plated on Todd Hewitt (TH) agar containing chloramphenicol (10 µg/ml) for an overnight incubation at 30 °C. Colonies were picked and transferred to a fresh plate and incubated at 45 °C for 2 days to select for integration of the plasmid into the chromosome. For double cross-over events, obtained single colonies were picked and incubated in TH broth without antibiotics with shaking at 200 rpm, 30 °C. Cultures were diluted 1:1000 and cultured for 8–16 h, resulting in 7 dilution-culturing cycles. The final culture was plated on TH agar containing 100 ng/mL anhydrotetracycline and incubated overnight at 37 °C to select for double cross-over mutants. Single colonies susceptible to chloramphenicol were analyzed via PCR for correct integration.

## Bacterial culture and inoculation

For inoculation with MW2 *S. aureus*, bacterial isolates were cultured in brain heart infusion medium (Becton Dickinson, Franklin Lakes, NJ, USA) overnight and adjusted to approximately $4.0 \times 10^6$ CFU in 20 µL medium. Bacterial suspensions (20 µL per ear pinna) were topically administered onto the ear pinnae of SPF mice. For the bacterial growth curve, 10 µL of bacterial isolates cultured in tryptic soy broth (TSB) (Becton Dickinson) overnight was added to 145 µL of pH-adjusted TSB and incubated with shaking at 200 rpm, 37 °C. To determine the bacterial growth rate, the OD (600 nm) was measured every 15 min using an infinite 200Pro with Tecan i-control (Tecan Japan Co., Ltd., Kawasaki, Japan).

## Antibiotic treatment

B6.VmC^SG1 mice received an antibiotic cocktail consisting of metronidazole (1 g/L), sulfamethoxazole (0.8 g/L), trimethoprim (0.16 g/L), cephalexin (4 g/L), and Baytril (0.025 g/L) dissolved in drinking water for two weeks to reduce skin microbial burden[49]. Baytril is enrofloxacin, which is more easily recognized as a DNA gyrase inhibitor in the fluoroquinolone class of antimicrobials. To ensure decreased microbial burden, cages were changed thrice weekly[50].

## Chemically induced atopic dermatitis-like disease

To induce a dermatitis-like state using MC903, the ear pinnae of 8-week-old B6 WT mice or B6.VmC^SG1 mice were treated daily with 1 nmol MC903 (calcipotriol, Tocris Bioscience, Bristol, UK) in 20 µL ethanol for 7 days (days 1–7). One day after the MC903 application, approximately $4.0 \times 10^6$ CFU of bacterial suspension in 20 µL medium was topically administered to the ear pinnae. For pH-adjusted buffer application, 40 µL of pH-adjusted 100 mM phosphate buffer was topically applied on the ear pinnae twice daily for four days. Ear thickness was measured with a digital micrometer (Mitutoyo, Kawasaki, Japan) at the end of each experiment.

## Protease activity

The activity of murine kallikrein-related peptidase (KLK)5 (R&D Systems, Minneapolis, MN, USA) and mesotrypsin (PRSS3) (R&D Systems) dissolved in various pH (3.5–8.0) buffers was measured according to the manufacturer's protocol. Briefly, a murine KLK5 activity assay was performed at 37 °C using 50 mM sodium phosphate buffer (pH

5.8–8.0) or 50 mM acetate buffer (pH 3.5–5.6) containing 0.50 µg/mL mouse KLK5 and 400 µM Boc-V-P-R-AMC Fluorogenic Peptide Substrate (R&D Systems) at the final concentrations. After 30 min of incubation, the fluorescence was measured at an excitation/emission wavelength of 380/460 nm using the PerkinElmer 2030 Multilabel Reader. The mouse PRSS3 activity assay was performed at 37 °C in 50 mM sodium phosphate buffer (pH 5.8–8.0) or 50 mM acetate buffer (pH 4.1–5.6) containing 0.02 µg/mL mouse PRSS3 and 50 µg/mL BODIPY FL casein (Thermo Fisher Scientific). After a 30-min incubation, the fluorescence was measured at an excitation/emission wavelength of 485/535 nm using the PerkinElmer 2030 Multilabel Reader. To determine the relative enzyme activity of murine KLK5 and PRSS3, relative enzyme activity was calculated based on the formula: $(A - B)/(C - B) \times 100$, where A = fluorescence at the indicated pH, B = fluorescence without enzymes, and C = maximum fluorescence obtained in the assay (i.e. pH 7.8 for murine KLK5 and pH 7.8 for PRSS3).

## Statistics

Data are expressed as the mean ± standard error of the mean. Differences among three or more groups were compared using Kruskal-Wallis one-way analysis of variance (ANOVA) followed by Dunn's post hoc tests, except in Fig. 5h, in which one-way ANOVA with Dunnett's comparison test was used. Dual comparisons were made using the two-sided Mann–Whitney test. Statistically significant differences are indicated as follows: $*p < 0.05$, $**p < 0.01$, $***p < 0.001$, $****p < 0.0001$. The mean values of single biological replicates in Fig. 1e, f, g, i, 2d, 3c, 4e, and Supplementary Fig. 5e are provided as Supplementary Data 1. GraphPad Prism 9 (GraphPad software, Boston, MA, USA) was used to perform analyses.

## Mathematical model of catalytic KLK activity in the SC

Our mathematical model explicitly describes four pH-dependent processes involved in the regulation of skin desquamation by active KLKs: (i) activation of pro-KLKs to KLKs, (ii) inhibition of KLKs by LEKTI, (iii) degradation of lymphoepithelial Kazal-type-related inhibitor (LEKTI), and (iv) KLK-mediated corneodesmosome proteolysis. These processes are summarized in a graphical scheme (Fig. 6a) and formulated using ordinary differential equations. We identified model functions and parameters from the relevant measurement data and used the established model to simulate the catalytic activity of KLKs under different hypothetical SC-pH profiles.

## Model development

We described the dynamics of the concentrations of pro-KLKs ($C_{pKLK}(x)$), KLKs ($C_{KLK}(x)$), LEKTI ($C_{LEKTI}(x)$), and the KLK–LEKTI complex ($C_{K-L}(x)$) at height $x$ in the SC:

$$\frac{dC_{pKLK}(x)}{dx} = -\frac{C_{pKLK}(x)}{v}\left[k_{auto} \cdot a_{KLK}(pH(x)) \cdot C_{KLK}(x) + k_{meso} \cdot a_{meso}(pH(x)) \cdot C_{meso}\right], \quad (1)$$

$$\frac{dC_{KLK}(x)}{dx} = \frac{1}{v}\{C_{pKLK}(x)\left[k_{auto} \cdot a_{KLK}(pH(x)) \cdot C_{KLK}(x) + k_{meso} \cdot a_{meso}(pH(x)) \cdot C_{meso}\right] \\ - k_a(pH(x)) \cdot C_{KLK}(x) \cdot C_{LEKTI}(x) + k_d(pH(x)) \cdot C_{K-L}(x) \\ + k_{deg} \cdot a_{meso}(pH(x)) \cdot C_{meso} \cdot C_{K-L}(x)\} \quad (2)$$

$$\frac{dC_{LEKTI}(x)}{dx} = \frac{1}{v}[-k_a(pH(x)) \cdot C_{KLK}(x) \cdot C_{LEKTI}(x) + k_d(pH(x)) \cdot C_{K-L}(x) \\ - k_{deg} \cdot a_{meso}(pH(x)) \cdot C_{meso} \cdot C_{LEKTI}(x)] \quad (3)$$

$$\frac{dC_{K-L}(x)}{dx} = \frac{1}{v}[k_a(pH(x)) \cdot C_{KLK}(x) \cdot C_{LEKTI}(x) - k_d(pH(x)) \cdot C_{K-L}(x) \\ - k_{deg} \cdot a_{meso}(pH(x)) \cdot C_{meso} \cdot C_{K-L}(x)] \quad (4)$$

**Table 1 | Model functions and parameters**

| Symbols | Explanation | Units |
|---|---|---|
| $pH(x)$ | pH at height $x$ (Fig. 6a) | - |
| $a_{KLK}(pH(x))$ | Relative catalytic activity of KLKs at each pH ($pH(x)$) to that at an optimal pH of 7.8 | - |
| $a_{meso}(pH(x))$ | Relative catalytic activity of mesotrypsin at each pH ($pH(x)$) to that at an optimal pH of 7.8 | - |
| $C_{meso}$ | Concentration of mesotrypsin in the SC | M |
| $C_{pKLK}^{SG/SC}$ | Concentration of pro-KLKs at $x = 0$ (SG/SC boundary) | M |
| $C_{LEKTI}^{SG/SC}$ | Concentration of LEKTI at $x = 0$ (SG/SC boundary) | M |
| $k_{auto}$ | Rate constant of pro-KLK autoactivation | M⁻¹s⁻¹ |
| $k_{meso}$ | Rate constant of pro-KLK activation via mesotrypsin | M⁻¹s⁻¹ |
| $k_a(pH(x))$ | Association rate constant of KLK and LEKTI to the KLK–LEKTI complex | M⁻¹s⁻¹ |
| $k_d(pH(x))$ | Dissociation rate constant of the KLK–LEKTI complex to KLK and LEKTI | s⁻¹ |
| $k_{deg}$ | Elimination rate constant of LEKTI via mesotrypsin | M⁻¹s⁻¹ |
| $v$ | Transit rate of pro-KLKs, KLKs, LEKTI, and the KLK–LEKTI complex from $x = 0$ (SG/SC boundary) to $x = 1$ (SC surface) | s⁻¹ |

*KLK* kallikrein-related peptidase, *LEKT1* lymphoepithelial Kazal-type-related inhibitor.

with the initial conditions $C_{pKLK}(0) = C_{pKLK}^{SG/SC}$, $C_{KLK}(0) = 0$, $C_{LEKTI}(0) = C_{LEKTI}^{SG/SC}$, and $C_{K-L}(0) = 0$. Pro-KLKs and LEKTI are secreted into the extracellular space between the SG and SC ($x = 0$) via the lamellar granule system, where LEKTI is separated from pro-KLKs[9,51,52].

There are no KLKs or KLK–LEKTI complexes at the SG/SC boundary, and pro-KLKs are activated into KLKs in the SC. Each term of the model equations is detailed under "Activation of pro-KLKs into KLKs," and the model functions and parameters are summarized in Table 1 and under "Inhibition of KLKs by LEKTI." Using this model, we evaluated the effective catalytic activity of KLKs at $x$ as

$$E_{KLK}(x) = a_{KLK}(pH(x)) \cdot C_{KLK}(x) \tag{5}$$

which is the relative catalytic activity of KLKs at each pH, compared with that at the optimal pH (pH 7.8).

### Derivation of ordinary differential equations
**Activation of pro-KLKs into KLKs.** Pro-KLKs are converted to KLKs by KLKs themselves (autoactivation) or by other proteases such as mesotrypsin[9]. We modeled this KLK activation process using the following equations:

$$\frac{dC_{pKLK}(x)}{dt} = -C_{pKLK}(x)\left[k_{auto} \cdot a_{KLK}(pH(x)) \cdot C_{KLK}(x) + k_{meso} \cdot a_{meso}(pH(x)) \cdot C_{meso}\right] \tag{6}$$

$$\frac{dC_{KLK}(x)}{dt}\bigg|_{1/3} = C_{pKLK}(x)\left[k_{auto} \cdot a_{KLK}(pH(x)) \cdot C_{KLK}(x) + k_{meso} \cdot a_{meso}(pH(x)) \cdot C_{meso}\right] \tag{7}$$

where $t$ denotes the time for pro-KLKs and LEKTI to reach height $x$ (from the time of their secretion at the SG/SC boundary ($x = 0$)) via corneocyte transit. The pH ($pH(x)$) depends on $x$ according to the pH profile (Fig. 6b–d). The catalytic activities of KLKs and mesotrypsin at each pH, $a_{KLK}(pH(x))$ and $a_{meso}(pH(x))$, are described by the relative activities when compared to their maximal activity at an optimal pH (pH 7.8). We assumed that mesotrypsin exists sufficiently across the entire SC to exert its catalytic activity because the influence of specific factors on the distribution of mesotrypsin has not yet been reported.

**Inhibition of KLKs by LEKTI.** LEKTI inhibits the catalytic activity of KLKs by reversibly forming a KLK–LEKTI complex that inactivates

KLKs. We modeled this process of KLK–LEKTI complex formation as:

$$\frac{dC_{KLK}(x)}{dt}\bigg|_{2/3} = -k_a(pH(x)) \cdot C_{KLK}(x) \cdot C_{LEKTI}(x) + k_d(pH(x)) \cdot C_{K-L}(x) \tag{8}$$

$$\frac{dC_{LEKTI}(x)}{dt}\bigg|_{1/2} = -k_a(pH(x)) \cdot C_{KLK}(x) \cdot C_{LEKTI}(x) + k_d(pH(x)) \cdot C_{K-L}(x) \tag{9}$$

$$\frac{dC_{K-L}(x)}{dt}\bigg|_{1/2} = k_a(pH(x)) \cdot C_{KLK}(x) \cdot C_{LEKTI}(x) - k_d(pH(x)) \cdot C_{K-L}(x) \tag{10}$$

Both the association and dissociation rates, $k_a(pH(x))$ and $k_d(pH(x))$, for the KLK–LEKTI complex are pH-dependent[38].

**LEKTI degradation.** Mesotrypsin degrades LEKTI both on its own and in KLK–LEKTI complexes[9]. This process of LEKTI degradation is modeled as:

$$\frac{dC_{KLK}(x)}{dt}\bigg|_{3/3} = k_{deg} \cdot a_{meso}(pH(x)) \cdot C_{meso} \cdot C_{K-L}(x) \tag{11}$$

$$\frac{dC_{LEKTI}(x)}{dt}\bigg|_{2/2} = -k_{deg} \cdot a_{meso}(pH(x)) \cdot C_{meso} \cdot C_{LEKTI}(x) \tag{12}$$

$$\frac{dC_{K-L}(x)}{dt}\bigg|_{2/2} = -k_{deg} \cdot a_{meso}(pH(x)) \cdot C_{meso} \cdot C_{K-L}(x) \tag{13}$$

We did not consider the degradation of pro-KLKs and KLKs because the influence of specific factors on the degradation of pro-KLKs and KLKs in the SC has not yet been reported.

**Corneocyte transit from the SG/SC boundary to the SC surface.** Corneocyte transit from the SG/SC boundary toward the SC surface occurs via epidermal turnover (Fig. 6a). This transit brings pro-KLKs, KLKs, LEKTI, and the KLK–LEKTI complex toward the SC surface. The time required for pro-KLKs and LEKTI to reach height $x$ from their secretion at the SG/SC boundary ($x = 0$) was defined by

$$t = \frac{x}{v} \tag{14}$$

where $v$ is the depth-independent transit rate of pro-KLKs, KLKs, LEKTI, and KLK–LEKTI.

**Dynamics of pro-KLKs, KLKs, LEKTI, and the KLK–LEKTI complex.** The dynamics of pro-KLK concentrations were described by a combination of Eqs. (6) and (14):

$$\frac{dC_{pKLK}(x)}{dx} = -\frac{C_{pKLK}(x)}{v}\left[k_{auto} \cdot a_{KLK}(pH(x)) \cdot C_{KLK}(x) + k_{meso} \cdot a_{meso}(pH(x)) \cdot C_{meso}\right] \tag{15}$$

The dynamics of KLK concentration were obtained by a combination of Eqs. (7), (8), (11), and (14):

$$\frac{dC_{KLK}(x)}{dx} = \frac{1}{v}\{C_{pKLK}(x)\left[k_{auto} \cdot a_{KLK}(pH(x)) \cdot C_{KLK}(x) + k_{meso} \cdot a_{meso}(pH(x)) \cdot C_{meso}\right]$$
$$-k_a(pH(x)) \cdot C_{KLK}(x) \cdot C_{LEKTI}(x) + k_d(pH(x)) \cdot C_{K-L}(x)$$
$$+ k_{deg} \cdot a_{meso}(pH(x)) \cdot C_{meso} \cdot C_{K-L}(x)\} \tag{16}$$

The dynamics of the LEKTI concentration were obtained by a combination of Eqs. (9), (12), and (14):

$$\frac{dC_{LEKTI}(x)}{dx} = \frac{1}{v}[-k_a(pH(x)) \cdot C_{KLK}(x) \cdot C_{LEKTI}(x) + k_d(pH(x)) \cdot C_{K-L}(x)$$
$$- k_{deg} \cdot a_{meso}(pH(x)) \cdot C_{meso} \cdot C_{LEKTI}(x)] \tag{17}$$

**Table 2 | KLK5 and LEKTI concentrations**

| Protein | Site of biopsy | Concentration | Ref. |
|---|---|---|---|
| KLK5 | Serum | $1.41 \times 10^{-11}$ M (391.07 pg/mL) | Wu et al.[48] |
| KLK5 | Stratum corneum | $1.12 \times 10^{-10}$ M* (3.1 ng/mg dry weight) | Komatsu et al.[49] |
| KLK5 | Epidermis | $2.61 \times 10^{-11}$ M* (0.86 ng/mg dry weight) | Fortugno et al.[50] |
| LEKTI (D10D15) | Epidermis | $1.46 \times 10^{-10}$ M* (9.48 ng/mg dry weight) | Fortugno et al.[50] |

*KLK* kallikrein-related peptidase, *LEKT1* lymphoepithelial Kazal-type-related inhibitor.

The dynamics of the KLK–LEKTI complex concentration were obtained by a combination of Eqs. (10), (13), and (14):

$$\frac{dC_{K-L}(x)}{dx} = \frac{1}{v}[k_a(pH(x)) \cdot C_{KLK}(x) \cdot C_{LEKTI}(x) \\ - k_d(pH(x)) \cdot C_{K-L}(x) - k_{deg} \cdot a_{meso}(pH(x)) \cdot C_{meso} \cdot C_{K-L}(x)]$$

(18)

**Model parameters and simulation method**

We derived the values for four parameters, $a_{KLK}(pH(x))$, $a_{meso}(pH(x))$, $k_a(pH(x))$, and $k_d(pH(x))$ by curve-fitting the in vitro experimental data (Supplementary Fig. 5a–c) and those for $C_{meso}$, $C_{pKLK}^{SG/SC}$, $C_{LEKTI}^{SG/SC}$, and $v$ from the literature.

We used three representative $k_{deg}$ values, each corresponding to one of the three scenarios: no degradation of LEKTI ($k_{deg} = 0$), continuous decreases in LEKTI ($k_{deg} = 2 \times 10^4$), and complete depletion of LEKTI ($k_{deg} = 2 \times 10^5$ at pH 7.0_5.4 [gradient]) and $3 \times 10^5$ at pH 7.0_6.0_5.4_6.7 (stepwise) and pH 7.0_5.4_6.7 (gradient). These three scenarios allowed us to deduce the effects of $k_{deg}$ on our results for the entire range of $k_{deg}$, but information on the distribution of LEKTI was lacking.

We identified the ranges of the remaining two parameters, $k_{auto}$ and $k_{meso}$, by evaluating the reproducibility of the expected spatial distribution of KLKs (detailed below under "Rates of pro-KLK activation via autoactivation ($k_{auto}$) and via mesotrypsin ($k_{meso}$)").

**pH-dependent catalytic activity of KLK: $a_{KLK}(pH(x))$.** We obtained the profile of $a_{KLK}(pH(x))$ by curve-fitting the relative activity of mouse KLK5 (a representative subtype of KLKs) at different pH values (Supplementary Fig. 5a). The relative activity was the ratio of the catalytic activity of KLK5 measured at each pH to that at pH 7.8 (optimal), and the catalytic activity values were obtained from the degradation rates of Boc-V-P-R-AMC Fluorogenic Peptide Substrate in vitro.

**pH-dependent catalytic activity of mesotrypsin: $a_{meso}(pH(x))$.** We obtained the profile of $a_{meso}(pH(x))$ by curve-fitting the relative activity of mouse mesotrypsin at different pH values (Supplementary Fig. 5b). The relative activity was the ratio of the catalytic activity of mesotrypsin measured at each pH to that at pH 7.8 (optimal), where the catalytic activity was obtained from the degradation rates of BODIPY FL casein in vitro.

**pH-dependent association and dissociation rates: $k_a(pH(x))$ and $k_d(pH(x))$.** Deraison et al. measured $k_a(pH(x))$ and $k_d(pH(x))$ at several pH values using surface plasmon resonance analysis[38]. Their values were used to obtain the profiles of $k_a(pH(x))$ and $k_d(pH(x))$ via curve-fitting (Supplementary Fig. 5c).

**Concentrations at the SG/SC boundary: $C_{pKLK}^{SG/SC}$ and $C_{LEKTI}^{SG/SC}$.** We assumed values for $C_{pKLK}^{SG/SC}$ and $C_{LEKTI}^{SG/SC}$ [$C_{pKLK}^{SG/SC}$, $C_{LEKTI}^{SG/SC}$] = [$1.0 \times 10^{-10}$, $1.0 \times 10^{-10}$] based on published measurement data[53–55] (Table 2).

**Mesotrypsin concentrations: $C_{meso}$.** We fixed the $C_{meso}$ values while tuning $k_{deg}$ and $k_{meso}$ in our simulation because $C_{meso}$ appears only in the forms of $k_{deg} \cdot C_{meso}$ or $k_{meso} \cdot C_{meso}$ in our model. We chose

$C_{meso} = 1.0 \times 10^{-10}$ to be of the same order as that of the concentrations of total KLKs ($C_{pKLK}(x) + C_{KLK}(x) + C_{K-L}(x) = C_{pKLK}^{SG/SC} = 1.0 \times 10^{-10}$). This allowed us to compare the contributions of KLK autoactivation versus mesotrypsin-mediated KLK activation ($-\frac{C_{pKLK}(x)}{v}\{k_{auto} \cdot a_{KLK}(pH(x)) \cdot C_{KLK}(x)\}$ and $-\frac{C_{pKLK}(x)}{v}\{k_{meso} \cdot a_{meso}(pH(x)) \cdot C_{meso}\}$) by comparing the values of $k_{meso}$ and $k_{auto}$.

**Transition rates of pro-KLKs, KLKs, LEKTI, and the KLK–LEKTI complex in the SC: $v$.** We assumed that proteins and corneocytes had the same transit time (the time for corneocytes to transit from the SG/SC boundary to the SC surface) in the SC and adopted $v = 2.3 \times 10^{-6}$ (/s) as the SC transit time of corneocytes in mice was measured to be 5 days ($1/[20 \text{ days} \times 24 \text{ h/day} \times 60 \text{ day/min} \times 60 \text{ min/s}] \simeq 2.3 \times 10^{-6}$/s)[56].

**Rate constant of mesotrypsin-mediated LEKTI elimination: $k_{deg}$.** The dynamics of the total concentration of LEKTI ($C_{Ltotal}(x) = C_{LEKTI}(x) + C_{K-L}(x)$) depend on $k_{deg}$ but are independent of KLK activation ($k_{auto}$ and $k_{meso}$), as described by a combination of Eqs. (17) and (18):

$$\frac{dC_{Ltotal}}{dx} = \frac{dC_{LEKTI}(x)}{dx} + \frac{dC_{K-L}(x)}{dx} \\ = \frac{1}{v}\{-k_{deg} \cdot a_{meso}(pH(x)) \cdot C_{meso} \cdot [C_{LEKTI}(x) + C_{K-L}(x)]\}$$

(19)

To obtain representative values of $k_{deg}$, we conducted model simulations using $k_{deg}$ values from 0 to $1.0 \times 10^6$ and picked the values of $k_{deg}$ that make $C_{Ltotal}(x)$ decrease to 100% (no degradation), 50% (continuous decrease), and 0.01% (complete depletion) at $x = 1$, that is, $\frac{C_{Ltotal}(1)}{C_{Ltotal}(0)} = 1, 0.5$, or $0.0001$, respectively. No degradation, continuous decreases, and complete depletion of LEKTI corresponded to the three example spatial distributions of LEKTI across the SC. These three scenarios would allow us to deduce the effects of $k_{deg}$ on our results for the entire range of $k_{deg}$, but information on the spatial distribution of LEKTI was lacking.

The selected values of $k_{deg}$ were 0, $2 \times 10^4$, and $3 \times 10^5$ for pH 7_5_7 (stepwise); 0, $2 \times 10^4$, and $2 \times 10^5$ for pH 7_5 (gradient); and 0, $2 \times 10^4$, and $3 \times 10^5$ for pH 7_5_7 (gradient) (Supplementary Fig. 5d). We confirmed that the model simulations using these $k_{deg}$ values exhibited no LEKTI degradation (solid lines in Supplementary Fig. 5e), continuous decreases in LEKTI (dashed lines), or complete LEKTI depletion (dotted lines), respectively.

**Rates of pro-KLK activation via autoactivation ($k_{auto}$) and via mesotrypsin ($k_{meso}$).** We identified the values of $k_{auto}$ and $k_{meso}$ that reproduced the expected dynamics for the total concentration of KLKs ($C_{Ktotal}(x) = C_{KLK}(x) + C_{K-L}(x)$). $C_{Ktotal}(x)$ is expected to increase throughout the SC because the close proximity (indicating interactions) between pro-KLKs and mesotrypsin throughout the SC[9] suggests that the activation of pro-KLKs to KLKs occurs throughout the SC, where activated KLK exists either on its own or within the KLK–LEKTI complexes. To reproduce this expected behavior, we chose values of $k_{deg}$, $k_{auto}$, and $k_{meso}$ that determined the dynamics of

$C_{\text{Ktotal}}(x)$, as described by a combination of Eq. (16) and (18):

$$\frac{dC_{\text{Ktotal}}(x)}{dx} = \frac{dC_{\text{KLK}}(x)}{dx} + \frac{dC_{\text{K-L}}(x)}{dx} = -\frac{dC_{\text{pKLK}}(x)}{dx}$$

$$= \frac{1}{v}\{C_{\text{pKLK}}(x)[k_{\text{auto}} \cdot a_{\text{KLK}}(pH(x)) \cdot C_{\text{KLK}}(x) + k_{\text{meso}} \cdot a_{\text{meso}}(pH(x)) \cdot C_{\text{meso}}]\}$$

$$(20)$$

Note that the dynamics of $C_{\text{KLK}}(x)$ depend on $k_{\text{deg}}$ (Eq. (2)).

Using the values of $k_{\text{deg}}$ determined (under "Rate constant of mesotrypsin-mediated LEKTI elimination: $k_{deg}$"), we performed model simulations for different values of $k_{\text{auto}}$ and $k_{\text{meso}}$ (both from $1.0 \times 10^2$ to $1.0 \times 10^6$) to identify example values of $k_{\text{auto}}$ and $k_{\text{meso}}$ that reproduced the increases in $C_{\text{Ktotal}}(x)$ throughout the SC. We specifically evaluated whether increases in $C_{\text{Ktotal}}(x)$ in the upper half of the SC ($0.5 < x < 1$) were comparable ($\pm 50\%$) to those in the bottom half of the SC ($0 < x < 0.5$) and whether $C_{\text{Ktotal}}(x)$ increased significantly (more than 50% of $C_{\text{pKLK}}(0)$, which is the maximal increase in $C_{\text{Ktotal}}(x)$) throughout the SC ($0 < x < 1$), that is,

$$0.5 < \frac{C_{\text{Ktotal}}(1) - C_{\text{Ktotal}}(0.5)}{C_{\text{Ktotal}}(0.5) - C_{\text{Ktotal}}(0)} < 1.5 \text{ and} \qquad (21)$$

$$0.5 < \frac{C_{\text{Ktotal}}(1)}{C_{\text{pKLK}}(0)} \qquad (22)$$

where $C_{\text{Ktotal}}(0) = 0$ and $C_{\text{pKLK}}(0) = C_{\text{pKLK}}^{\text{SG/SC}}$.

Our model simulation revealed that certain values of $k_{\text{auto}}$ and $k_{\text{meso}}$ satisfied the conditions in Eqs. (21) and (22) for each $k_{\text{deg}}$ value (Supplementary Fig. 5f, g, and 6a). We found that four combinations of $k_{\text{auto}}$ and $k_{\text{meso}}$ values, which corresponded to the edges and center of the identified areas for the condition in Eqs. (21) and (22), showed continuous increases in $C_{\text{Ktotal}}(x)$ throughout the SC ($0 < x < 1$) (Supplementary Fig. 6a). However, combinations of $k_{\text{auto}}$ and $k_{\text{meso}}$ values that did not satisfy the conditions in Eqs. (21) and (22) failed to increase $C_{\text{Ktotal}}(x)$ (Supplementary Fig. 6a) or activated most of the pro-KLKs in the bottom half of the SC, leaving only minimal activation of pro-KLKs in the upper half of the SC (Supplementary Fig. 6a).

**Model simulation.** We simulated the catalytic activity of KLKs in murine SC using hypothetical pH profiles (Fig. 6b–d) to evaluate the biological importance of stepwise pH profiles in murine SC. To test whether our models with different pH gradients could reproduce the increases in $E_{\text{KLK}}(x)$ toward the SC surface, we performed a simulation for the range of $k_{\text{auto}}$ and $k_{\text{meso}}$ values identified above (under "Rates of pro-KLK activation via autoactivation ($k_{auto}$) and via mesotrypsin ($k_{meso}$)") and the three example values for $k_{\text{deg}}$ determined above (under "Rate constant of mesotrypsin-mediated LEKTI elimination: $k_{deg}$"). We evaluated the increases in $E_{\text{KLK}}(x)$ from $x = 0.7$ to $x = 1.0$, i.e., $E_{\text{KLK}}(1.0) - E_{\text{KLK}}(0.7)$ (Supplementary Fig. 6b). In addition, we simulated the catalytic activity of KLKs in uniform pH 5.4 and pH 7.0 models, respectively (Supplementary Fig. 6c). To interpret the influence of disrupted pH profiles on SC homeostasis, we performed a simulation of our model using $k_{deg}$, $k_{\text{auto}}$, and $k_{\text{meso}}$ values within the parameter ranges identified above (under "Rates of pro-KLK activation via autoactivation ($k_{auto}$) and via mesotrypsin ($k_{meso}$)") and the three example values for $k_{\text{deg}}$ determined above (under "Rate constant of mesotrypsin-mediated LEKTI elimination: $k_{deg}$").

**Reporting summary**

Further information on research design is available in the Nature Portfolio Reporting Summary linked to this article.

## Data availability

The raw data generated in this study to plot graphs and uncropped Western blots are provided in the Source Data file. Source data are provided with this paper.

## Code availability

All the codes[57] for mathematical modeling were written, and the simulations were conducted in Python version 3.8.1. The source code to reproduce the presented results is available at the online code repository: https://github.com/Tanaka-Group/stepwise_pH (DOI identifier: https://github.com/Tanaka-Group/pH_gradient).

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

## Acknowledgements

We thank R. Ozawa for assisting with the generation of B6.VmC^SG1, B6.VH148GmC^SG1, and B6.Display-VmC^SG1 mice, R. Yokoo for isolating SG1 cells and assisting confocal imaging analysis of SG1 cells, Hachiro Iseki for electron microscopy imaging, and M. Furuse for sharing the *Cldn1^{+/-}* frozen mouse embryo. This project was supported by Japan Society for the Promotion of Science (JSPS) Grant-in-Aid for Scientific Research (S) (22H04994 awarded to M.A.), JSPS Grant-in-Aid for Innovative Areas (15H05948 awarded to A.M), JSPS Grant-in-Aid for Scientific Research (C) (21K08356 awarded to K.F.) and Japan Agency for Medical Research and Development (AMED) (JP21ek0410058 and 21gm1010001 awarded to M.A.).

## Author contributions

K.F., Y.I., Y.F., T.Ma., A.M., and M.A. designed the study. K.F., Y.I., Y.F., and T.Ma. performed the experiments. H.H., T.Mi., M.V.L., and R.J.T. performed the mathematical modeling analysis. T.O. assisted with confocal microscopy techniques. A.M. provided VenusH148G-mCherry construct, assisted with image analysis, and provided feedback on the

manuscript. K.F., Y.I., and M.A. wrote the manuscript, and all authors read and edited the manuscript.

## Competing interests

The authors declare no competing interests.

## Ethics

All animal experiments were performed in accordance with the Ethical Guidelines of the Animal Care and Use Committee of the RIKEN Yokohama Institute (AEY2021-018(2)), Keio University Institutional Animal Care and Use Committee (A2022-285), and ARRIVE guidelines.
