## [Peer Review File · Nature Communications]

REVIEWER COMMENTS

Reviewer #1 (Remarks to the Author):

This meticulous study makes several key advances in our understanding of the cutaneous barrier, which is essential to terrestrial survival, immune surveillance, and infection prevention. The current work leverages innovative biochemical, imaging, gnotobiotic, and computer modeling approaches to elucidate how stratum corneum (SC) pH regulation contributes to skin barrier function and how it is disrupted by external pH insults and tight junction (TJ) dysfunction. Interestingly, the authors find that normal skin flora actually contribute to (or even directly stimulate) SC pH regulation to permit normal colonization within the outermost neutral SC layers. But SC cells prevent deeper microbial penetration by establishing an acidic middle SC layer, long known as the “acid mantle” of the skin. The authors go on to show that dysbiosis from *Staph aureus*, a common cutaneous pathogen thought to drive atopic dermatitis (AD), in combination with chemically induced atopic-like inflammation, results in disruption of the acidic SC layers, which permits deeper bacterial invasion. Finally, the authors establish computational models to explain how the maintenance of these distinct pH zones in the SC is also critical for titrating the enzymatic activity of KLKs, which are essential to tightly regulate the timing of protease activities that drive desquamation and preserve proper SC thickness.

Questions to be addressed by the authors via reply to reviewers and manuscript text updates:

1) Should we refer to SC cells as “post-mortem?” The authors nicely show that these cells seem to actively regulate their pH and KLK enzyme activities well after nuclear degradation. Mature erythrocytes, which are similarly anucleate and lack mitochondria, remain metabolically active and are not thought of as being dead.

2) I am curious to know how the authors propose that “post-mortem” SC cells accomplish the task defined in this paper. In the Discussion, can the authors shed light on their proposed mechanism of pH regulation in the SC layers? Is there active pumping of protons to maintain pH? Is the process ATP-dependent? Mitochondria are known to be disassembled in the SG layers, so I presume Ox-Phos is not happening in the SC layers.

3) How is pH regulation altered as SC cells transition from one layer to the next? In prior work from this group, the TRPV3 channel was found to be up-regulated to control calcium influx in the SG layers. Is there a similar mechanism in the SC to regulate pH? Beyond transcription (since the nucleus is gone), how would a channel/pump be differentially regulated in SC layers to maintain the distinct pH zones?

Questions warranting additional clarifying experiments if possible and/or explanations:

Fig. 2e, f: The contribution of FLG to skin barrier function is supported by genetic studies showing its mutation causes ichthyosis vulgaris and confers susceptibility to atopic dermatitis. What does FLG staining look like in the *Cldn1* KO mice? Does the loss of the pH zones directly impact expression or processing of FLG? Can tissue cross-section staining similar to Fig. 2a be done on KO skin?

Fig. 3b: Does alkaline buffer application similarly disrupt the pH zones? Perhaps this experiment is not compatible with the available pH-sensitive intravital fluorophores, but the experiments in Fig. 5 use alkaline buffers in conjunction with MC903 +/- *Staph aureus*. I am curious what the alkaline buffer treatment alone does to the SC pH zones. This is clinically relevant since dilute bleach baths (which are alkaline, pH ~9) are thought to improve skin barrier function in AD; while this may be due to microbial de-colonization (particularly of *Staph aureus*), hypochlorite has also been shown to have direct effects on epidermal keratinocytes (Leung et al. *J Clin Invest* 2013 Dec;123(12):5361-70; doi: 10.1172/JCI70895).

Fig. 4a, b: As above, I am left wondering what would be the effect of dilute hypochlorite, which is used to treat AD. One might predict this would eliminate microbes similar to the broad antibiotic regimen used, but hypochlorite being alkaline might actually disrupt the middle SC acid mantle layer? Or perhaps the presence of TJs would still preserve the acidic layer--could the authors clarify?

Fig. 5d: Studies of the role of FLG in AD have suggested bi-directional influences between the cutaneous barrier and the immune system. While FLG deficiency compromises the barrier and can promote skin inflammation, others propose that intrinsic inflammation, itself, down-regulates FLG to compromise the barrier. Does MC903 affect FLG expression or breakdown?

Fig. 5h: Why does the pH 3.5 buffer seem to be an outlier in the graph? The authors propose the acidic middle SC layer serves to prevent deeper cutaneous microbial growth. However, it seems the pH 3.5 buffer yielded a comparable increase in ear thickness as pH 8.2 and pH 10.3, which is surprising. Also, how would the authors reconcile their results with the positive effect of dilute hypochlorite (bleach) baths, on reducing *Staph aureus* colonization and improving AD as shown in clinical trials? Are the alkaline buffers used in these experiments antimicrobial?

Minor Issues:

Fig. 4c: Please provide more details about the SPF mice. Which specific pathogens are they free of? I assume they are known to be free of *Staph aureus*? Is it known which flora they DO have present on the skin that permit the normal 3 pH zones of the SC?

Fig. 6c: I believe there is an error showing “Active KLK” in the uppermost layer of the right-hand diagram. The enzyme is colored blue and I think it should be labeled “Inactive KLK.”

In the methods section, the authors should specify that Baytril is enrofloxacin, which is more easily recognized as a DNA gyrase inhibitor in the fluoroquinolone class of antimicrobial.

Reviewer #2 (Remarks to the Author):

The dead cornified layer of the skin epidermis is a major component of the skin epidermal barrier function. The stratum corneum with its corneocytes is the final step in a terminal differentiation of keratinocytes. The importance of a properly functioning cornified barrier is underscored by human diseases in which this terminal differentiation process is disturbed, with improper cornification and hyperkeratinization. In addition, despite enucleation corneocytes are actively shed through cleavage of corneodesmosomes, indicating that biochemically this layer is still active. However, what regulates barrier properties of this zone and how terminal differentiation controls formation and homeostasis of this layer is unclear. Through work among others from these authors, we know that generation specific pH milieus in the SC compartment is important to among others spatially regulate the activity of proteolytic enzymes necessary for lipid and protein processing and desquamation. However, technically it has been challenging to determine pH in vivo and assess how changing the pH may alter barrier function and terminal differentiation. In this elegant, well written manuscript the authors have met this challenge and unravel a stepwise regulation of the pH throughout the SC by precisely calculating the exact pH using different fluorescent reporters, revealing the existence of 3 stepwise spatially separated pH zones. Through a combination of challenging the pH zones with different treatments, the use of epidermal knockout mice as well as altering the microbiome, the authors provide some evidence for how these zones are generated. They also assign differential function/properties for each of these zones that they can link to either the regulation of the pH or to changes in the pH, which are important for barrier formation, and also in part explain spatiotemporal regulation of proteolytic enzymes. They then use an elegant modelling approach integrating both pH-dependent regulation of enzyme activity and inhibitor protein regulation of activity to show how a three-step zone of pH changes promote proper desquamation. (the reviewer is not an expert on modelling and cannot judge the modelling and simulation data). This study addresses a fundamental biological process (can enucleated so called dead cells actually still undergo further differentiation to generate functionally different layers of “dead cells”?) with very high clinical relevance. The experiments are in general quite comprehensive and the data shown are of high quality. Only a few points should be addressed.

Although next to the claudin-1 knockout, it would be favorable to see a somewhat more comprehensive crossing in of these reporters in other genetic backgrounds to see how changes in key barrier components that serve as models for human skin diseases affect the pH zonation, this would be outside

of the scope of the current manuscript. Having said this, the role of terminal differentiation and pH zone would be interesting to explore more within this manuscript. In other words if one would for example affect final processing of filaggrin using caspase 14 inhibition or inhibit kalikrin activity from outside first affecting the most outer barrier layer, would that affect pH and/or turnover? The latter would also help to provide additional evidence for the modelling that they present.

The second point relates somewhat to the first one. Although I like their modelling that indicates that a three step pH model would promote desquamation, this model is not really put to any test, neither experimentally nor in silico. The paper would gain in significance when added. For example, would changing the three zones or their pH directly affects their processing/enzyme activity in vivo as predicted? The analysis in the main figures is by and large based on IF, but it would be nice when one disturbs the pH of the zones using chemicals or zonation itself (the claudin-1 KO mice, which shows altered SC and perhaps desquamation, see below) does this alter filaggrin processing or KLK activity and thus corneodesmin cleavage? This could be done in combination with tape stripping like they did for control in supplementary figure 2.

Vice versa, the authors could address whether the pH changes observed in the claudine ok mice impair desquamation in the model, as that model in the initial in vivo description seem to suggest altered desquamation (Furuse et al. 2002).

Minor points:

If not mistaken, the strategy of HR.SASPVENUS-mCherry/+ mice generates a heterozygous SASP knockout. The earlier publication of Matsui et al. 2011 on the SASP knockout indicates that the heterozygous do not have a phenotype. However, this should be mentioned in the manuscript as this may be less obvious for some readers.

Images in figure 1 suggest that there is little to no mCherry fluorescence in upper SC which cannot be explained by the pH. This effect does not seem to be consistent throughout other figures e.g. Fig.3a. mcCherry fluorescence should be rather equal throughout layers as also suggested by the experiments in Fig. S1b. How do the authors explain this loss of fluorescence in vivo and ex vivo and how do they incorporate when determining the mCherry-Venus ratio or calculating the exact pH?

How do differential photobleaching properties affect the mcherry to venus ratio. Did the authors acquire confocal stack in top-down and bottom-up to compare how that affects ratios? The ratios were determined in isolated cells that do not require big stacks.

The Venus-mCherry reporter if not released to the extracellular space during corneoptosis acts as a intracellular pH-sensor whereas proteolytic activity at corneodesmosomes leading to desquamation depends on extracellular corneodesmosome cleavage and thus extracellular pH levels would determine

desquamation. Is there evidence of extracellular localization of the sensor or do intracellular pH levels match the extracellular milieu in the SC due to increased membrane permeability during corneoptosis?

In various experiments single cell measurements are pooled from different mice and pooled values are used for statistical analyses using regular ANOVA and post test. This is not ideal as the biological variation is somewhat hidden through high numbers of single cells. Hence more stringent tests should be use e.g. Kruskal-Wallis followed by Dunn's or else. Otherwise the p-values will suggest too high significance. In addition the mean values of single biological replicates should be shown in the supplementary data to show the biological variation.

Specific points:

Figure1.

Figure1 b,d,e,f,g Unlike the other figures Fluorescent images are not labeled with the fluorophores and the colors they are shown in.

e-g if single cell measurements are pooled from 3 different mice and if pooled values are used for statistical analyses, a more stringent ANOVA should be use e.g. Kruskal-Wallis followed by Dunn's or else.

Reviewer #3 (Remarks to the Author):

This manuscript by Fukuda et al probes the relationship between the intracellular pH of corneocytes and SC maturation in vivo. Using quantitative intravital pH imaging of transgenic mice that accumulate genetically encoded ratiometric pH indicators in corneocytes, their work uncovered the segmentation of the stratum corneum into three zones with distinct pH values –with an expected near neutral pH for corneocytes near the SC surface. Using claudin-1 KO mice, they further show that tight junctions are required for this SC zonation. Inoculations and mathematical modeling uncovered further functional implications of this three-tiered zonation towards protection against *S. aureus* colonization as well as regulation of kallikrein-related peptidase family activity for desquamation. The emerging new framework is distinct from the common notion of a neutral-to-acidic profile of the stratum corneum –as observed in prior work with tools largely restricted to the extracellular space of the SC. Their tools and findings will stimulate the functional dissection of how corneocytes mature to sustain the life-defining skin barrier. In the spirit of strengthening the manuscript for a broad audience, I encourage the authors to consider the following moderate (M) and minor (Mi) points:

(M1) The use of “postmortem” in the title masks a major strength of the paper: that they pioneer intracellular pH measurements in the SC in vivo –in living mice! I strongly encourage the authors to consider an updated title. Rather than “postmortem”, their findings challenge the view of the SC as dead (“postmortem”) tissue. The carefully orchestrated pH zonation is a new important demonstration of the complex intracellular and tissue level maturation of corneocytes that enables SC homeostasis. Using “postmortem” in the title is a missed opportunity to present their new findings to a broad audience.

(M2) Fukuda et al demonstrate conserved pH zonation across different body sites in mice, but the manuscript lacks data (or at the very least discussion) of how their findings extend to human skin –which would be of interest to a broad audience. For example, human skin equivalents are readily accessible but do not exhibit desquamation. Whether this defect is partly rooted at the level of SC intracellular pH zonation deserves consideration/discussion.

(M3) Related to (2), the manuscript would benefit from strengthening the discussion of their new data in relation to prior measurements of SC pH gradient. In particular, the manuscript does not distinguish between intracellular pH and extracellular pH. From my point of view, the novelty of the new work stems from developing tools to pioneer the probing of the intracellular pH of corneocytes. Prior approaches have relied on tools that are largely restricted to extracellular spaces in the SC –like dyes with limited SC penetration. The current manuscript does not seem to properly discuss these important differences. If properly discussed, however, the authors will gain (1) greater emphasis on the novelty of their tools/approach, and (2) a proper integration of their new findings into the extant literature on the pH of the SC –like the well-known extracellular acidic pH of the SC surface.

Minor points:

(Mi-1) The statement "lower SC that expresses filaggrin" is inaccurate, suggesting active transcription/expression in corneocytes. The manuscript would benefit from referring to "lower SC with filaggrin peptides". In general, using language throughout the text that marks an accurate distinction between FLG and filaggrin (peptides) would be helpful, avoiding the use of "expression" for the observation of filaggrin peptides.

(Mi-2) Figure 1d is labeled twice. Figure 1c label is missing.

Responses to Reviewers

Reviewers' comments are itemized below in italicized 12-point font.

The authors' responses to each specific comment are in blue 12-point font with text line numbers in red.

Reviewer #1:

*This meticulous study makes several key advances in our understanding of the cutaneous barrier, which is essential to terrestrial survival, immune surveillance, and infection prevention. The current work leverages innovative biochemical, imaging, gnotobiotic, and computer modeling approaches to elucidate how stratum corneum (SC) pH regulation contributes to skin barrier function and how it is disrupted by external pH insults and tight junction (TJ) dysfunction. Interestingly, the authors find that normal skin flora actually contribute to (or even directly stimulate) SC pH regulation to permit normal colonization within the outermost neutral SC layers. But SC cells prevent deeper microbial penetration by establishing an acidic middle SC layer, long known as the “acid mantle” of the skin. The authors go on to show that dysbiosis from *Staph aureus*, a common cutaneous pathogen thought to drive atopic dermatitis (AD), in combination with chemically induced atopic-like inflammation, results in disruption of the acidic SC layers, which permits deeper bacterial invasion. Finally, the authors establish computational models to explain how the maintenance of these distinct pH zones in the SC is also critical for titrating the enzymatic activity of KLKs, which are essential to tightly regulate the timing of protease activities that drive desquamation and preserve proper SC thickness.*

Questions to be addressed by the authors via reply to reviewers and manuscript text updates:

1) Should we refer to SC cells as “post-mortem?” The authors nicely show that these cells seem to actively regulate their pH and KLK enzyme activities well after nuclear degradation. Mature erythrocytes, which are similarly anucleate and lack mitochondria, remain metabolically active and are not thought of as being dead.

1) Thank you for your suggestion. We agree that the word “post-mortem” in the title is misleading. As suggested, our findings revise the view of the SC as an inactive or unresponsive dead cell layer. Therefore, we changed the title of our paper to “**Three stepwise pH progressions in stratum corneum for homeostatic maintenance of the skin.**” We also removed the word “post-mortem” from the manuscript.

2) I am curious to know how the authors propose that “post-mortem” SC cells accomplish the task defined in this paper. In the Discussion, can the authors shed light on their proposed mechanism of pH regulation in the SC layers? Is there active pumping of protons to maintain pH? Is the process ATP-dependent? Mitochondria are known to be disassembled in the SG layers, so I presume Ox-Phos is not happening in the SC layers.

2) A recent proteome study showed that peptides of vacuolar-type ATPase, which pumps protons across the plasma membranes using ATP hydrolysis energy, can be detected in healthy human SC (Dyring-Andersen B, et al. Nat Commun. 2020 Nov 5;11(1):5587). However, as the reviewer pointed out, mitochondria do not exist in corneocytes, and ATP synthesis by oxidative phosphorylation is unlikely to occur in the SC. Thus, it is unclear whether vacuolar-type ATPase

is active in the SC to maintain SC-pH. These considerations were added to the Discussion section (Page 17, lines 18-23).

3) How is pH regulation altered as SC cells transition from one layer to the next? In prior work from this group, the TRPV3 channel was found to be up-regulated to control calcium influx in the SG layers. Is there a similar mechanism in the SC to regulate pH? Beyond transcription (since the nucleus is gone), how would a channel/pump be differentially regulated in SC layers to maintain the distinct pH zones?

3) Currently, the mechanism of three-stepwise SC-pH progression is unknown. We agree that this is one of the most critical issues to be determined. In particular, elucidation of the molecular mechanism of the acidic middle SC layer formation will contribute to the development of a new therapeutic strategy for improving skin barrier functions.

Cryo-electron microscopy (cryo-EM) revealed that the organization of the lipid lamellae drastically changes in the intercellular spaces between the first and the fifth stratum corneum cells of the SC: (1) a single-band pattern with 2.0–2.5 nm periodicity and (2) a two-band pattern with 5.5–6.0 nm periodicity (Narangifard, A, et al. J Invest Dermatol. 2021 May;141(5):1243-1253). Since the optimal pH of the lipid metabolism enzyme is acidic, we speculate that the lipid metabolism enzymes that contribute to the formation of lipid lamellae organization regulate SC-pH. These considerations were added to the Discussion section (Page 17, lines 23 - Page 18, lines 4).

Questions warranting additional clarifying experiments if possible and/or explanations:

*Fig. 2e, f: The contribution of FLG to skin barrier function is supported by genetic studies showing its mutation causes ichthyosis vulgaris and confers susceptibility to atopic dermatitis. What does FLG staining look like in the *Cldn1* KO mice? Does the loss of the pH zones directly impact expression or processing of FLG? Can tissue cross-section staining similar to Fig. 2a be done on KO skin?*

We thank the reviewer for pointing out these important issues. We examined the SC-pH of epidermis-specific *Cldn1* KO VmC^{SG1} mice (B6.*Cldn1*^{fl/fl} K14-CreERT VmC^{SG1} mice). Similar to neonatal *Cldn1*^{-/-} VH148GmC^{SG1} mice, the entire SC of adult B6.*Cldn1*^{fl/fl} K14-CreERT VmC^{SG1} mice exhibited a neutral pH. In addition, we performed immunofluorescence staining of FLG in the skin of B6.*Cldn1*^{fl/fl} K14-CreERT VmC^{SG1} mice. We found that FLG was expressed with accumulation along the cell membranes of corneocytes throughout the SC; however, the western blot analyses of SC samples obtained by tape stripping revealed that there was no band for FLG monomer. Instead, we detected bands of dimetric, trimetric, and tetrameric FLGs. Collectively, these results suggest that a deficiency of claudin-1 resulted in the loss of three stepwise SC-pH pH zones and the accumulation of prematurely processed FLGs; this led to the decrease of FLG monomer and natural moisturizing factor production. Data are shown in Fig. 2g and Supplementary Figure 3d-f, and are described in the Results section (Page 9, lines 22 - Page 10, lines 11) of the revised manuscript.

Fig. 3b: Does alkaline buffer application similarly disrupt the pH zones? Perhaps this experiment is not compatible with the available pH-sensitive intravital fluorophores, but the experiments in Fig. 5 use alkaline buffers in conjunction with MC903 +/- Staph aureus. I am curious what the

alkaline buffer treatment alone does to the SC pH zones. This is clinically relevant since dilute bleach baths (which are alkaline, pH ~9) are thought to improve skin barrier function in AD; while this may be due to microbial de-colonization (particularly of Staph aureus), hypochlorite has also been shown to have direct effects on epidermal keratinocytes (Leung et al. J Clin Invest 2013 Dec;123(12):5361-70; doi: 10.1172/JCI70895).

Yes, this is an important point. MC903-induced dermatitis is known to induce dysbiosis (Amar Y, et al. J Eur Acad Dermatol Venereol. 2022 May;36(5):705-716). Our mice treated with MC903 + *S. aureus* were in SPF conditions, not gnotobiotic conditions. Thus, the fluorescence *S. aureus* application would not significantly affect SC-pH. Therefore, we would expect that alkaline buffer application neutralizes the middle SC acid mantle layer in MC903-applied skin, similar to the results of mice treated with MC903 + *S. aureus* + alkaline buffer.

Fig. 4a, b: As above, I am left wondering what would be the effect of dilute hypochlorite, which is used to treat AD. One might predict this would eliminate microbes similar to the broad antibiotic regimen used, but hypochlorite being alkaline might actually disrupt the middle SC acid mantle layer? Or perhaps the presence of TJs would still preserve the acidic layer--could the authors clarify?

This is also an important point to be addressed. To answer this question clearly, we need to measure the pH of the middle SC-pH zone, TJ function by permeability assay using different molecular weight tracer, and skin microbiota composition by 16S rRNA gene sequencing. Therefore, we feel that this is beyond the scope of this manuscript. In future investigations, we would like to pursue this issue.

Fig. 5d: Studies of the role of FLG in AD have suggested bi-directional influences between the cutaneous barrier and the immune system. While FLG deficiency compromises the barrier and can promote skin inflammation, others propose that intrinsic inflammation, itself, down-regulates FLG to compromise the barrier. Does MC903 affect FLG expression or breakdown?

The expression of FLG protein is down-regulated in MC903-induced dermatitis (Nasanbat B, et al. J Dermatol Sci. 2023 Sep;111(3):93-100). Furthermore, IL-4 and IL-13 inhibit the expression of FLG protein by activating STAT6 and STAT3 (Furue M. Int J Mol Sci. 2020 Jul 29;21(15):5382). We assume that the activation of STAT6 and STAT3 led to the downregulation of FLG protein expression in our MC903-induced dermatitis model since we previously demonstrated that MC903-induced dermatitis induces IL-4 and IL-13 (Ito Y, et al. Cell Rep. 2021 Apr 27;35(4):109052).

Fig. 5h: Why does the pH 3.5 buffer seem to be an outlier in the graph? The authors propose the acidic middle SC layer serves to prevent deeper cutaneous microbial growth. However, it seems the pH 3.5 buffer yielded a comparable increase in ear thickness as pH 8.2 and pH 10.3, which is surprising. Also, how would the authors reconcile their results with the positive effect of dilute hypochlorite (bleach) baths, on reducing Staph aureus colonization and improving AD as shown in clinical trials? Are the alkaline buffers used in these experiments antimicrobial?

Thank you for the question. We do not have a clear answer to this question. As shown in **Supplementary Figure. 4g**, the alkaline buffers used in these experiments are not antimicrobial.

We observed that pH 3.5 buffer significantly suppressed the KLK catalytic activity compared with the pH 6.6 buffer, and mice that received the pH 3.5 buffer exhibited a thicker SC than mice that received the pH 6.6 buffer (Supplementary Figure 6d-f). Therefore, we speculate that suppressing KLK catalytic led to increased ear thickness.

Minor Issues:

Fig. 4c: Please provide more details about the SPF mice. Which specific pathogens are they free of? I assume they are known to be free of *Staph aureus*?

SPF mice are free of *S. aureus* in our institution. We added this information in the Results section in Page 12, lines 5-6. In addition, SPF conditions are free from *Corynebacterium kutscheri*, *Mycoplasma pulmonis*, *Salmonella spp.*, *Clostridium piliforme*, Ectromelia Virus, LCM virus, Mouse hepatitis virus, Sendai virus, Ectoparasites, Intestinal protozoa, Pinworm, *Citrobacter rodentium*, *Pasteurella pneumotropica*, and *Helicobater hepaticus*.

Is it known which flora they DO have present on the skin that permit the normal 3 pH zones of the SC?

Yes, this is an important point to be addressed. We have yet to develop a fluorescent protein labeled for bacteria other than *S. aureus* and do not have the answer to this question at this moment. Although the pH of the upper SC-pH zone of germ-free mice was significantly lower than that of SPF mice, germ-free mice exhibited lower-moderately acidic and middle-acidic SC-pH zone. In addition, germ-free mice co-housed with SPF mice displayed a neutralized upper SC with similar pH values to SPF mice. Collectively, these findings indicate that skin microbiota is important for the neutralization of the upper SC-pH zone and that factors other than skin microbiota are required for the development of three stepwise SC-pH zones.

Fig. 6c: I believe there is an error showing "Active KLK" in the uppermost layer of the right-hand diagram. The enzyme is colored blue and I think it should be labeled "Inactive KLK."

Thank you. We changed "Active KLK" to "Inactive KLK".

In the methods section, the authors should specify that Baytril is enrofloxacin, which is more easily recognized as a DNA gyrase inhibitor in the fluoroquinolone class of antimicrobial.

Thank you. We added this information in the Methods section (Page 27, lines 5-6).

Reviewer #2 (Remarks to the Author):

The dead cornified layer of the skin epidermis is a major component of the skin epidermal barrier function. The stratum corneum with its corneocytes is the final step in a terminal differentiation of keratinocytes. The importance of a properly functioning cornified barrier is underscored by human diseases in which this terminal differentiation process is disturbed, with improper cornification and hyperkeratinization. In addition, despite enucleation corneocytes are actively shed through cleavage of corneodesmosomes, indicating that biochemically this layer is still active. However, what regulates barrier properties of this zone and how terminal differentiation controls formation and homeostasis of this layer is unclear. Through work among others from these authors, we know that generation specific pH milieu in the SC compartment is important

to among others spatially regulate the activity of proteolytic enzymes necessary for lipid and protein processing and desquamation. However, technically it has been challenging to determine pH *in vivo* and assess how changing the pH may alter barrier function and terminal differentiation. In this elegant, well written manuscript the authors have met this challenge and unravel a stepwise regulation of the pH throughout the SC by precisely calculating the exact pH using different fluorescent reporters, revealing the existence of 3 stepwise spatially separated pH zones. Through a combination of challenging the pH zones with different treatments, the use of epidermal knockout mice as well as altering the microbiome, the authors provide some evidence for how these zones are generated. They also assign differential function/properties for each of these zones that they can link to either the regulation of the pH or to changes in the pH, which are important for barrier formation, and also in part explain spatiotemporal regulation of proteolytic enzymes. They then use an elegant modelling approach integrating both pH-dependent regulation of enzyme activity and inhibitor protein regulation of activity to show how a three-step zone of pH changes promote proper desquamation. (the reviewer is not an expert on modelling and cannot judge the modelling and simulation data). This study addresses a fundamental biological process (can enucleated so called dead cells actually still undergo further differentiation to generate functionally different layers of "dead cells"?) with very high clinical relevance. The experiments are in general quite comprehensive and the data shown are of high quality. Only a few points should be addressed.

(1) Although next to the claudin-1 knockout, it would be favorable to see a somewhat more comprehensive crossing in of these reporters in other genetic backgrounds to see how changes in key barrier components that serve as models for human skin diseases affect the pH zonation, this would be outside of the scope of the current manuscript. Having said this, the role of terminal differentiation and pH zone would be interesting to explore more within this manuscript. In other words, if one would for example affect final processing of filaggrin using caspase 14 inhibition or inhibit kallikrein activity from outside first affecting the most outer barrier layer, would that affect pH and/or turnover? The latter would also help to provide additional evidence for the modelling that they present.

(2) The second point relates somewhat to the first one. Although I like their modelling that indicates that a three step pH model would promote desquamation, this model is not really put to any test, neither experimentally nor *in silico*. The paper would gain in significance when added. For example, would changing the three zones or their pH directly affects their processing/enzyme activity *in vivo* as predicted?

(1) (2) We thank the reviewer for pointing out these important issues. To examine the impact of suppressing KLK catalytic activity in the upper SC-pH zone on desquamation *in vivo*, we used the pH 3.5 buffer, which significantly suppresses the catalytic activity of mouse KLK compared with a pH 6.6 buffer, on the ear of B6.VmC^{SG1} mice for four consecutive days (Supplementary Fig. 6d). The mice that received the pH 3.5 buffer showed thicker SC compared to the mice that received the pH 6.6 buffer (Supplementary Fig. 6e). These results suggest that suppression of KLK catalytic activity leads to the suppression of desquamation, thereby increasing SC thickening. These results were compatible with those obtained using our mathematical model. Data are shown in Supplementary Fig. 6d,e and are described in the Results section (Page 15, lines 10-17) of the revised manuscript.

The analysis in the main figures is by and large based on IF, but it would be nice when one disturbs the pH of the zones using chemicals or zonation itself (the claudin-1 KO mice, which shows altered SC and perhaps desquamation, see below) does this alter filaggrin processing or KLK activity and thus corneodesmin cleavage? This could be done in combination with tape stripping like they did for control in supplementary figure 2. Vice versa, the authors could address whether the pH changes observed in the claudin ko mice impair desquamation in the model, as that model in the initial in vivo description seem to suggest altered desquamation (Furuse et al. 2002).

Thank you for the suggestion. This question was also raised by reviewer 1. We examined the SC-pH of epidermis-specific *Cldn1* KO VmC^{SG1} mice (B6.*Cldn1*^{fl/fl} K14-CreERT VmC^{SG1} mice). Similar to neonatal *Cldn1*^{-/-} VH148GmC^{SG1} mice, the entire SC of adult B6.*Cldn1*^{fl/fl} K14-CreERT VmC^{SG1} mice exhibited a neutral pH and thicker SC compared to VmC^{SG1} mice. In addition, we performed immunofluorescence staining of FLG in the skin of B6.*Cldn1*^{fl/fl} K14-CreERT VmC^{SG1} mice. We found that FLG distributed with accumulation along the cell membranes of corneocytes in the entire SC; however, western blot analyses of SC samples obtained by tape stripping revealed that there was no expression of FLG monomer. Instead, we detected bands of dimetric, trimetric, and tetrameric FLGs. Collectively, these results suggest that a deficiency of claudin-1 in the epidermis resulted in the loss of three stepwise SC-pH zones and the accumulation of prematurely processed filaggrin. These TJ-impaired conditions are considered to result in decreased filaggrin monomer and suppression of subsequent natural moisturizing factor production, which would lead to impaired desquamation. Data are shown in Fig. 2g and Supplementary Figure 3d-f, and are described in the Results section (Page 9, lines 22 - Page 10, lines 11) of the revised manuscript.

Minor points:

If not mistaken, the strategy of HR.SASPVENUS-mCherry/+ mice generates a heterozygous SASP knockout. The earlier publication of Matsui et al. 2011 on the SASP knockout indicates that the heterozygous do not have a phenotype. However, this should be mentioned in the manuscript as this may be less obvious for some readers.

Thank you for your suggestion. Yes, we previously reported that heterozygous SASP KO does not have a phenotype (Matsui et al. 2011). Thus, we added the information that HR.SASPVENUS-mCherry/+ mice generate a heterozygous SASP knockout and that heterozygous SASP KO does not have a phenotype in the Result sections (Page 5, lines 17-19) of the revised manuscript.

Images in figure 1 suggest that there is little to no mCherry fluorescence in upper SC which cannot be explained by the pH. This effect does not seem to be consistent throughout other figures e.g. Fig.3a. mCherry fluorescence should be rather equal throughout layers as also suggested by the experiments in Fig. S1b. How do the authors explain this loss of fluorescence in vivo and ex vivo and how do they incorporate when determining the mCherry-Venus ratio or calculating the exact pH?

We appreciate the reviewer for highlighting this important issue. We believe this is a technical issue and does not change our conclusion. During pH calibration using isolated SG1 cells, we had to set the laser power of mCherry lower than Venus (e.g., 3.70 μ W for Venus and 2.88 μ W for

mCherry) to prevent potential overexposure of mCherry fluorescence. This calibration setting was maintained for intravital SC-pH imaging to ensure consistency. In this setting, mCherry signals tended to represent weak fluorescent in the upper SC-pH zone; however, the dedicated mCherry image still confirmed the presence of mCherry (figure below), and we were able to accurately calculate the SC-pH.

How do differential photobleaching properties affect the mcherry to venus ratio. Did the authors acquire confocal stack in top-down and bottom-up to compare how that affects ratios? The ratios were determined in isolated cells that do not require big stacks.

We appreciate this important issue. We are aware that photobleaching properties affect the mcherry-to-venus ratio. To overcome this issue, we captured a picture of the XY-plane image and examined the groove where we could simultaneously observe the SC layer without needing stack multiple pictures.

In regard to pH calibration using isolated SG1 cells, XY-planes were scanned at a speed of 400 Hz using a laser with its power set at (i) 3.70 μW for Venus and 2.88 μW for mCherry or (ii) 3.42 μW for Venus and 2.53 μW for mCherry (the same setting was used for intravital SC-pH imaging), and 30 XY-plane images with a 0.50- μm Z spacing were constructed. Images were then stacked using Fiji software's "MAX intensity" script. Using this method, we did not find the difference in mcherry-to-venus ratio whether the SG1 cells were scanned in the top-down or bottom-up mode, as shown in the figure below.

The Venus-mCherry reporter if not released to the extracellular space during corneoptosis acts as an intracellular pH-sensor whereas proteolytic activity at corneodesmosomes leading to desquamation depends on extracellular corneodesmosome cleavage and thus extracellular pH levels would determine desquamation. Is there evidence of extracellular localization of the sensor or do intracellular pH levels match the extracellular milieu in the SC due to increased membrane

permeability during corneoptosis?

We appreciate this important issue. This question was also raised by Reviewer 3. To examine whether the extracellular pH of corneocytes show similar pH changes, we generated B6 mice that specifically express Venus-mCherry fusion protein on the extracellular side of the plasma membrane of SG1 cells and corneocytes (B6.SASP^{Display-Venus-mCherry/+} mice [B6. Display-VmC^{SG1} mice]) and performed intravital SC-pH imaging. Similar to VmC^{SG1} and VH148GmC^{SG1} mice, B6. Display-VmC^{SG1} mice exhibited three-tiered zonation. The upper, middle, and lower SC-pH zones and the SG1 cell layer had significantly different pH values: pH 6.76 (95% CI, 6.61–6.91), 5.36 (95% CI, 5.30–5.41), 5.82 (95% CI, 5.76–5.88), and 6.80 (95% CI, 6.67–6.93), respectively. These results imply that the extracellular pH of corneocytes is consistent with that of the intracellular pH of corneocytes. Data are shown in Fig. 1h,i, and Supplementary Movie 2, and are described in the Results sections (Page 7, lines 8-18) of the revised manuscript.

In various experiments single cell measurements are pooled from different mice and pooled values are used for statistical analyses using regular ANOVA and post test. This is not ideal as the biological variation is somewhat hidden through high numbers of single cells. Hence more stringent tests should be use e.g. Kruskal-Wallis followed by Dunn's or else. Otherwise the p-values will suggest too high significance. In addition the mean values of single biological replicates should be shown in the supplementary data to show the biological variation.

Thank you for your suggestion. We re-analyzed data of Fig.1e-g, 3c, and 4e, where we used one-way ANOVA with Tukey's multiple-comparisons test, by performing Kruskal-Wallis one-way ANOVA followed by Dunn's post hoc tests. Although the p-values changed, the conclusion did not change in all the experiments. In addition, we created the Supplementary Table 1, which shows the mean values of single biological replicates.

Specific points:

Figure1.

Figure1 b,d,e,f,g Unlike the other figures Fluorescent images are not labeled with the fluorophores and the colors they are shown in.

Thank you. We labeled the fluorescent images in Figure 1b,d,e,f, and g with the fluorophores and colors.

e-g if single cell measurements are pooled from 3 different mice and if pooled values are used for statistical analyses, a more stringent ANOVA should be use e.g. Kruskal-Wallis followed by Dunn's or else.

Thank you. We re-analyzed Fig.1e-g data by performing Kruskal-Wallis one-way ANOVA followed by Dunn's post hoc tests. Although the p-values changed, the conclusion did not change in all the experiments.

Reviewer #3 (Remarks to the Author):

This manuscript by Fukuda et al probes the relationship between the intracellular pH of corneocytes and SC maturation in vivo. Using quantitative intravital pH imaging of transgenic mice that accumulate genetically encoded ratiometric pH indicators in corneocytes, their work uncovered the segmentation of the stratum corneum into three zones with distinct pH values – with an expected near neutral pH for corneocytes near the SC surface. Using claudin-1 KO mice,

they further show that tight junctions are required for this SC zonation. Inoculations and mathematical modeling uncovered further functional implications of this three-tiered zonation towards protection against *S. aureus* colonization as well as regulation of kallikrein-related peptidase family activity for desquamation. The emerging new framework is distinct from the common notion of a neutral-to-acidic profile of the stratum corneum –as observed in prior work with tools largely restricted to the extracellular space of the SC. Their tools and findings will stimulate the functional dissection of how corneocytes mature to sustain the life-defining skin barrier. In the spirit of strengthening the manuscript for a broad audience, I encourage the authors to consider the following moderate (M) and minor (Mi) points:

(M1) The use of “postmortem” in the title masks a major strength of the paper: that they pioneer intracellular pH measurements in the SC *in vivo* –in living mice! I strongly encourage the authors to consider an updated title. Rather than “postmortem”, their findings challenge the view of the SC as dead (“postmortem”) tissue. The carefully orchestrated pH zonation is a new important demonstration of the complex intracellular and tissue level maturation of corneocytes that enables SC homeostasis. Using “postmortem” in the title is a missed opportunity to present their new findings to a broad audience.

(M1) Thank you for your suggestion. This question was also raised by Reviewer 1. We agree that “post-mortem” in the title is misleading. As suggested, our findings revise the view of the SC as an inactive or unresponsive dead cell layer. We changed the title of our paper to **“Three stepwise pH progressions in stratum corneum for homeostatic maintenance of the skin”**. We also removed the word “post-mortem” from the manuscript.

(M2) Fukuda *et al* demonstrate conserved pH zonation across different body sites in mice, but the manuscript lacks data (or at the very least discussion) of how their findings extend to human skin –which would be of interest to a broad audience. For example, human skin equivalents are readily accessible but do not exhibit desquamation. Whether this defect is partly rooted at the level of SC intracellular pH zonation deserves consideration/discussion.

(M2) We appreciate this important issue. We are also keen to know the SC-pH distribution in the human skin and performed intracellular SC-pH imaging of *in vitro* 3D-human skin equivalent (HSE) stably expressing Venus-mCherry fusion protein after retroviral transduction. Unfortunately, we found that the stratum corneum of HSE lacked three stepwise pH zones, and the whole SC showed neutral pH (Figure below).

Since we found that the SC of *Cldn1*^{-/-} mice lacks the three stepwise SC-pH zones and the whole SC exhibited neutral pH, we speculate that HSEs have functionally-non-matured TJ function,

thereby showing neutral pH as *Cldn1*^{-/-} mice. Indeed, several reports demonstrate that HSEs exhibit higher permeability than native human skin, suggesting impaired skin barrier functions (Bouwstra JA, et al. *Adv Drug Deliv Rev.* 2021 Aug;175:113802). Thus, we think it would require intradermal injection or topical application of Venus-mCherry containing plasmid DNA to the human skin explant to examine SC-pH in human skin. We would like to address this issue as an immediate next project, which may take at least a year or so, but could not include it in this revision. We hope that the reviewer will understand the current situation.

(M3) Related to (2), the manuscript would benefit from strengthening the discussion of their new data in relation to prior measurements of SC pH gradient. In particular, the manuscript does not distinguish between intracellular pH and extracellular pH. From my point of view, the novelty of the new work stems from developing tools to pioneer the probing of the intracellular pH of corneocytes. Prior approaches have relied on tools that are largely restricted to extracellular spaces in the SC –like dyes with limited SC penetration. The current manuscript does not seem to properly discuss these important differences. If properly discussed, however, the authors will gain (1) greater emphasis on the novelty of their tools/approach, and (2) a proper integration of their new findings into the extant literature on the pH of the SC –like the well-known extracellular acidic pH of the SC surface.

(M3) We agree, and corrected the text in the Discussion section stating that we developed a tool that enables intravital intracellular SC-pH imaging, whereas previous tools mainly measured the extracellular pH of corneocytes (Page 16, lines 3-6).

Furthermore, we generated B6 mice that specifically express Venus-mCherry fusion protein on the extracellular side of the plasma membrane of SG1 cells and corneocytes (B6.SASP^{Display-Venus-mCherry/+} mice [B6. Display-VmC^{SG1} mice]) and performed intravital SC-pH imaging to examine whether the extracellular pH of corneocytes shows similar pH changes, like VmC^{SG1} and VH148GmC^{SG1} mice. B6. Display-VmC^{SG1} mice exhibited three-tiered zonation. The upper, middle, and lower SC-pH zones and the SG1 cell layer had significantly different pH values: pH 6.76 (95% CI, 6.61–6.91), 5.36 (95% CI, 5.30–5.41), 5.82 (95% CI, 5.76–5.88), and 6.80 (95% CI, 6.67–6.93), respectively. These results imply that the extracellular pH of corneocytes is consistent with that of the intracellular pH of corneocytes. Data are shown in Fig. 1h,i, and Supplementary Movie 2, and are described in the Results sections (Page 7, lines 8-18) of the revised manuscript.

Minor points:

(Mi-1) The statement "lower SC that expresses filaggrin" is inaccurate, suggesting active transcription/expression in corneocytes. The manuscript would benefit from referring to "lower SC with filaggrin peptides". In general, using language throughout the text that marks an accurate distinction between FLG and filaggrin (peptides) would be helpful, avoiding the use of "expression" for the observation of filaggrin peptides.

(Mi-1) Thank you for your suggestion. We changed the statement "lower SC that expresses filaggrin" to "lower SC with filaggrin" (Page 16, lines 24 - Page 17, lines 1). We also avoided using "expression" for the observation of filaggrin.

(Mi-2) Figure 1d is labeled twice. Figure 1c label is missing.

(Mi-2) Thank you. We changed "Figure 1d" to "Figure 1c."

REVIEWERS' COMMENTS

Reviewer #1 (Remarks to the Author):

In this very carefully executed and insightful study, the authors significantly advance our understanding of stratum corneum (SC) biology, which is critical to skin barrier function. Additional work in this revised version using distinct mice with a reporter to confirm the EXTRAcellular pH of the SC zones was particularly impressive and further confirms the original conclusions. As well, the added data showing disruption of pH zonation in claudin-deficient mice impairs processing of filaggrin further substantiates the key role of pH regulation in SC barrier function. The authors have adequately addressed my original concerns and/or provided sufficient acknowledgement of reasonable remaining limitations in the manuscript. I look forward to seeing this work published.

Reviewer #2 (Remarks to the Author):

The authors have addressed my most important comments and even if not all of my questions could be addressed this manuscript is a very important contribution to the field and it would be important to see this published in Nature Communications.

Reviewer #2 (Remarks on code availability):

I am not an expert in this area and feel that I cannot sufficiently judge this.

Reviewer #3 (Remarks to the Author):

I thank the authors for their efforts to address my suggestions and those of others reviewers. In particular, the new title and the addition of key data (namely the clever experimental effort to measure extracellular pH) greatly strengthen the manuscript. Their tools to measure both intracellular and extracellular pH in the SC are cutting-edge and provide fresh light into the stepwise functional maturation of the SC. I don't have any additional concerns and am very excited about their tools and findings.